mathematical modelling/health and disease and epidemiology

epidemic modelling, pandemic modelling, COVID-19, SARS-CoV-2, higher education

**Authors for correspondence:**
Jessica Enright
e-mail: jessica.enright@glasgow.ac.uk
Edward M. Hill
e-mail: Edward.Hill@warwick.ac.uk
Julia R. Gog
e-mail: jrg20@cam.ac.uk
Michael J. Tildesley
e-mail: M.J.Tildesley@warwick.ac.uk

†These authors contributed equally to this work.

# SARS-CoV-2 infection in UK university students: lessons from September–December 2020 and modelling insights for future student return

Jessica Enright[1,†], Edward M. Hill[2,3,†], Helena B. Stage[3,4], Kirsty J. Bolton[5], Emily J. Nixon[3,6], Emma L. Fairbanks[5,7], Maria L. Tang[3,8], Ellen Brooks-Pollock[3,9], Louise Dyson[2,3], Chris J. Budd[10], Rebecca B. Hoyle[11], Lars Schewe[12], Julia R. Gog[3,8] and Michael J. Tildesley[2,3]

[1]School of Computing Science, University of Glasgow, Glasgow G12 8QQ, UK
[2]The Zeeman Institute for Systems Biology and Infectious Disease Epidemiology Research, School of Life Sciences and Mathematics Institute, University of Warwick, Coventry CV4 7AL, UK
[3]Joint UNIversities Pandemic and Epidemiological Research, https://maths.org/juniper/
[4]Department of Mathematics, The University of Manchester, Oxford Road, Manchester, UK
[5]Centre for Mathematical Medicine and Biology, School of Mathematical Sciences, University of Nottingham, University Park, Nottingham, UK
[6]Veterinary Public Health, Bristol Veterinary School, University of Bristol, Bristol, UK
[7]School of Veterinary Medicine and Science, University of Nottingham, Loughborough, UK
[8]Department of Applied Mathematics and Theoretical Physics, University of Cambridge, Cambridge, UK
[9]Bristol Medical School, Population Health Sciences, University of Bristol, Bristol, UK
[10]School of Mathematical Sciences, University of Bath, Claverton Down, Bath, UK
[11]School of Mathematical Sciences, University of Southampton, Southampton, UK
[12]University of Edinburgh, School of Mathematics, James Clerk Maxwell Building, Peter Guthrie Tait Road, Edinburgh, UK

JE, 0000-0002-0266-3292; EMH, 0000-0002-2992-2004; HBS, 0000-0001-9938-8452; KJB, 0000-0003-0487-4701; EJN, 0000-0002-1626-9296; ELF, 0000-0002-1598-962X; MLT, 0000-0002-9671-8302; EB-P, 0000-0002-5984-4932; LD, 0000-0001-9788-4858; CJB, 0000-0003-4536-1662; RBH, 0000-0002-1645-1071; LS, 0000-0002-3778-262X; JRG, 0000-0003-1240-7214; MJT, 0000-0002-6875-7232

In this paper, we present work on SARS-CoV-2 transmission in UK higher education settings using multiple approaches to assess the extent of university outbreaks, how much those outbreaks may have led to spillover in the community, and the expected effects of control measures. Firstly, we found that the distribution of outbreaks in universities in late 2020 was consistent with the expected importation of infection

from arriving students. Considering outbreaks at one university, larger halls of residence posed higher risks for transmission. The dynamics of transmission from university outbreaks to wider communities is complex, and while sometimes spillover does occur, occasionally even large outbreaks do not give any detectable signal of spillover to the local population. Secondly, we explored proposed control measures for reopening and keeping open universities. We found the proposal of staggering the return of students to university residence is of limited value in terms of reducing transmission. We show that student adherence to testing and self-isolation is likely to be much more important for reducing transmission during term time. Finally, we explored strategies for testing students in the context of a more transmissible variant and found that frequent testing would be necessary to prevent a major outbreak.

# 1. Introduction

The global spread of SARS-CoV-2 has resulted in widespread usage of social distancing measures and non-pharmaceutical interventions (NPIs) to inhibit the spread of infection. Enactment of nationwide lockdowns has resulted in the closure of workplaces, pubs and restaurants, restricted leisure activities and impacted the education sector.

Measures brought in when entering the first nationwide lockdown in the UK in March 2020 included closure of higher education establishments, such as universities, to most in-person activities. Face-to-face teaching was mostly suspended, with delivery of the remainder of the 2019/2020 academic year taking place via online delivery.

Higher education in the UK comprises a large population of students, with over 2.3 million higher education students enrolled in the 2018/2019 academic year across over 160 higher education providers [1] (universities, essentially). This results in a sizeable movement of students nationwide at the beginning and end of academic terms (in addition to international student travel). In the context of an ongoing disease outbreak, the migration of students can contribute to increased population mobility, with an associated need for careful management in order to minimize the risk of seeding outbreaks both in universities and in the wider community.

Ahead of the 2020/2021 academic year, there was significant uncertainty around whether students would be able to return to face-to-face teaching and what policies would be put in place in order to mitigate risk. This prompted action to build a foundation of knowledge such that appropriate policies could be put in place to facilitate students returning safely to universities. From 15 to 17 June 2020, a Virtual Study Group on 'Unlocking Higher Education Spaces' was hosted by the Virtual Forum for Knowledge Exchange in the Mathematical Sciences (V-KEMS), looking at how mathematical approaches could inform the reopening of higher education spaces to students while minimizing risk. A working paper was subsequently released in July 2020 [2].

Building on the discussion that took place at the June 2020 Study Group, two virtual events (taking place on 28 July 2020 and 4 August 2020, respectively) investigated the application of mathematical tools and models to various issues linked to the challenges of reopening higher education. These events were run as part of the Isaac Newton Institute Infectious Dynamics of Pandemics Research Programme [3]. After these events, a working group continued to meet virtually on a weekly basis, consisting of participants from several institutions.

Mathematical modelling approaches informed by data, have been a valuable tool used to inform policy decisions linked to the subsequent operation of higher education in the midst of a pandemic. In order to guide these decisions, in this paper, we have investigated contributing factors to within-institution spread and how transmission interplays with the wider community. This study starts with a set of observational analyses based on data from the first term of the 2020/2021 academic year. This is followed by prospective modelling of control measures that were under consideration for the full return of UK higher education students in January 2021.

The work presented in this paper is the outcome of bringing together the expertise from these multiple research groups, and pooling our analyses using both statistical and modelling methods. Several conclusions emerge from this work both in understanding the observations from Autumn 2020, and also making recommendations for future actions:

(1) The overall distribution of outbreaks in universities in autumn term 2020 were consistent with expected importations from taking a student intake from the wider community, so that universities reflect the community disease prevalence at the start of term.

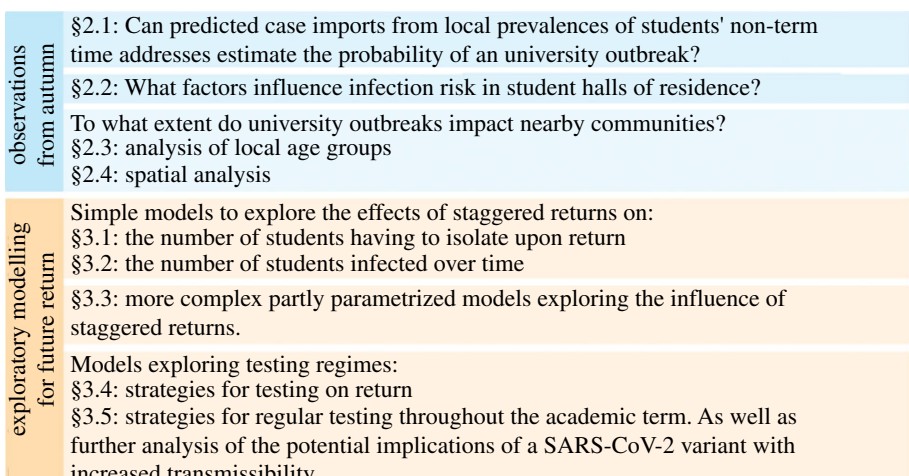

**Figure 1.** Overview of the structure of the article.

(2) Larger halls of residence pose higher risks for larger attack rates, and segmentation into smaller households within halls is unlikely to be able to mitigate this.

(3) The picture of transmission from universities to their local communities is complex. While spillover inevitably can occur, sometimes even large outbreaks in universities do not give any corresponding signal in their wider neighbouring communities.

(4) The proposed strategy of staggering future returns appears to be of somewhat mixed and limited value. While it could reduce the need for self-isolation on return under low prevalence, these benefits could be diminished or even reversed in the context of high background prevalence.

(5) While a staggered return could reduce the peak of any outbreak during term, staggering on its own will not substantially reduce the total attack rate over a whole term: staggering may act mainly to delay the outbreak to later in the term.

(6) The level of student adherence to testing and isolation is likely to have a far larger effect than any subtleties between different staggered return regimes.

(7) While it is likely that asymptomatic testing programmes did help to prevent large outbreaks in university settings in autumn 2020, extremely frequent testing (every 3 days) would be needed to prevent a major outbreak under plausible parameters for the B.1.1.7 variant (WHO variant label 'Alpha').

The structure of the remainder of this paper is as follows. In §2, we summarize the understanding and learning from the observed patterns of SARS-CoV-2 from autumn term 2020, looking at the dynamics of wider community transmission including the importation of cases to universities at the start of term, and the spillover of transmissions from universities to the wider community during the course of the term. This section also looks at the dependence on the infection dynamics within universities of the structures of halls of residence and student households. In §3, we look at several exploratory models for the future return of students, in particular looking at the impact of different strategies for staggering this return, and of asymptomatic testing on return. In §4, we draw some further conclusions from this work and make some policy recommendations. For a more detailed overview, see figure 1.

## 2. Observations from autumn term 2020

Higher education institutions in the UK largely reopened to students for the 2020/2021 academic year. This led to an influx of students from across the UK and world, brought together in residential, academic and social settings. In the first term, under the government advice at the time [4], most higher education establishments offered blended online and face-to-face learning. Prior to the beginning of the academic year, students resident in housing of multiple occupancy—and in particular students in residential halls—were identified as being at high risk of transmitting SARS-CoV-2 infection [5,6].

The return of students to universities in the autumn term occurred at a time when SARS-CoV-2 cases were growing in the UK. Local lockdowns came into force in areas with greatest risk leading to an increase in restrictions on travel, business openings and between-household socializing. In addition,

countrywide lockdowns were imposed in Wales from 23 October 2020 to 9 November 2020 and in England from 5 November 2020 to 2 December 2020. Many universities offered testing regimes in an attempt to further control outbreaks. In an attempt to segment interactions and reduce transmission risk within halls, many universities assigned students in residential halls to households based on the use of shared facilities such as kitchens and bathrooms under government guidance [4]. These households were intended to function similarly to households in the community; with many restrictions on socializing beyond these household members, and requirements for the entire household to isolate for up to 14 days if a member displayed symptoms of COVID-19 or received a positive SARS-CoV-2 test. Despite the control measures taken, outbreaks of varying sizes were seen in many UK higher education institutions in the first term, prompting concern about the possibility of spillover into the community.

In this section, we use data from the first term of the 2020/2021 academic year to investigate the factors that may have contributed to the observed outbreaks within higher education institutions and to examine any evidence of further transmission between higher education institutions and the wider community. Firstly, we consider the mass migration of students from across the UK at the beginning of term and how well this may explain the occurrence of outbreaks seen across universities (§2.1). We then use data available from a particular university and investigate the role of accommodation structure upon transmission, by considering the relationship of residential hall sizes and household sizes within halls to attack rates (§2.2). To investigate spillover from higher education to the community, we investigate case data by age (henceforth 'age-stratified') from areas very close to English universities to determine whether there is any evidence of spillover from student age groups to other age groups (§2.3). We also consider total case data stratified across a wider spatial scale to search for signs of spillover from areas with a high concentration of student residents to geographically nearby areas without high concentrations of students (§2.4).

## 2.1. Start of term: transmission from the community

Although many universities experienced outbreaks at the beginning of the 2020/2021 academic year, there was significant variation in the number of confirmed SARS-CoV-2 cases between institutions. We explore the extent to which the estimated incoming numbers of infected students could explain the observed distribution of outbreaks in the early weeks of the autumn term across UK universities.

### 2.1.1. Data and methods

To estimate the number of incoming infected students for each university at the beginning of the 2020/2021 academic year, we combined Office for National Statistics (ONS) infection survey data on the proportion of the community testing positive (prevalence) via polymerase chain reaction (PCR) to SARS-CoV-2 by region with data from the higher education Statistics Agency (HESA) on home and term-time postcodes for the 2018/2019 cohort of students [7]. The prevalence (via PCR) on 25 September 2020 was used to estimate the number of students from each home postcode that were infected at the start of term (rounded to the nearest integer). It was assumed that international students from countries with high case numbers would be placed in effective quarantine and were thus discounted for the purpose of this analysis. Outbreak data were drawn from the University and College Union (UCU) dashboard in November 2020 [8]. After omitting data with obvious quality issues, data for 72 universities were available. We defined a large outbreak as 200 or more cumulative cases reported on the UCU dashboard by 18 or 19 November 2020 (these case numbers obtained relate to various dates in November since updates were not daily or uniform).

To estimate the probability of a large outbreak, universities were binned by the estimated number of PCR-positive students in bin widths of 10, and the fraction of universities in each bin that experienced an outbreak was calculated based on the observed data.

We also considered a simple probabilistic model for the outbreak probability $\mathcal{P}$ based only on incoming PCR-positive students, $\mathcal{P} = 1 - p^n$, where $n$ corresponds to the initial number of PCR-positive students, and the extinction probability, $p$, is the probability that an incoming infection fails to seed an outbreak. The probabilities of each incoming infection seeding an outbreak are assumed to be independent of each other. The extinction probability, $p$, was inferred via maximum likelihood from the observed outbreak data (see appendix A).

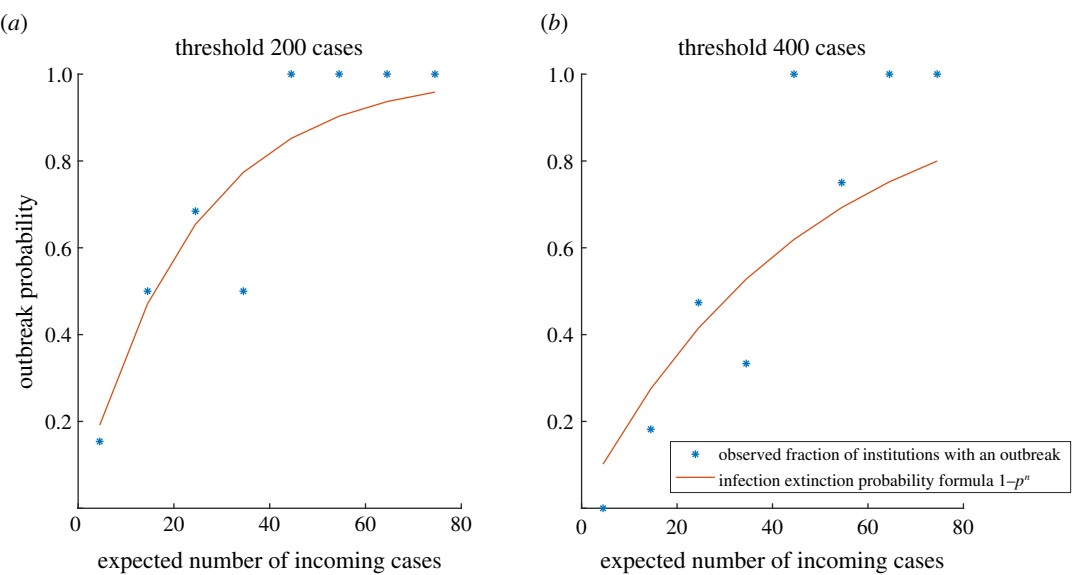

**Figure 2.** Observed fraction of institutions having an outbreak (*), binned by expected number of incoming cases, and theoretical outbreak probability $\mathcal{P}$ (solid line): for a threshold of 200 cases (*a*) and 400 cases (*b*).

### 2.1.2. Results

The observed fraction of universities experiencing an outbreak appeared to be broadly consistent with the simple probabilistic model (figure 2), with a fitted extinction probability of $p = 0.958$ (95% confidence interval [0.945, 0.972]). Repeating the analysis and fitting the simple model using a more stringent threshold of 400 cases returned an extinction probability estimate of $p = 0.979$ (95% confidence interval [0.971, 0.987]), with the model estimations following the trend of the observed data. These results lend cautious support to the hypothesis that the observed pattern of outbreaks at universities was consistent with that expected from importation of infection from the student intake.

This would imply that outbreaks are more likely when case numbers in the incoming student population are higher (higher $n$ leads to higher outbreak probability $\mathcal{P}$). Similarly, if the extinction probability, $p$, i.e. the probability of the chain of infection originating from a single introduction dying out, were lower then the overall outbreak probability $\mathcal{P}$ would be higher. Less effective infection control measures or a more transmissible variant might lead to a lower $p$, but this needs to be investigated further.

### 2.1.3. Limitations

Factors that we did not take into account in this simple initial analysis and that could be explored further include: the detailed timeline of importations and onward transmissions, the likelihood that an outbreak might be the sum of smaller outbreaks caused by independent introductions, the rate of assimilation of local prevalence in newly arrived students, the impact of heterogeneous university characteristics (such as the number of commuting students), and the impact of heterogeneity in university infection control measures.

In addition, we were limited by the availability of data; ideally the analysis should be repeated with contemporary student numbers and home regions, and with more consistent data on university case numbers.

In light of these limitations, the precise numerical value of the fitted extinction probability should not be interpreted literally. However, the fact that the extinction probability appears to be high suggests that the majority of infection chains die out before sparking an outbreak. This may be partly because COVID-19 is highly overdispersed [9] so that only a small proportion of infections lead to further cases, while many people with the disease do not infect anyone else. It may also reflect effective infection control measures in universities, or that there were fewer incoming infections than assumed in the model, perhaps because students who were unwell may have delayed their return to university,

or because estimates of the prevalence via PCR-testing includes people who are in the late stages of infection and no longer infectious.

## 2.2. Infection risk in residential student halls

Prior to the resumption of the 2020/2021 academic year there was limited data to relate transmission risk within halls and their households to that estimated for community households. Here, we examine factors predicting risk of infection among students in halls of residence at a single university. We refer to the secondary attack rate (SAR) in a subpopulation (e.g. household, hall of residence) as the probability that a member of the subpopulation is infected following infection of one subpopulation member.

### 2.2.1. Data

Data on hall capacity for 19 halls managed by the university, and the assignment of rooms within these into households of up to 16 members, were collected prior to the start of term. Stock data on room types for each hall was used to estimate the fraction of students sharing bathroom facilities with at least one other student for each hall and in each household. During term, students were encouraged to report confirmed SARS-CoV-2 infection via a web form, including information about their place of residence, date of test result and subject. Preliminary enrolment data for 2020/2021 by subject and term-time residence were used to estimate the fraction of students in each hall enrolled in the Medical Faculty (as a proxy for students who may be at higher risk of infection due to placements). Approximately half of students reported a room number in addition to identifying their hall of residence, which enabled these reported infections to be grouped into pre-assigned households of known size.

### 2.2.2. Methods

We tested for predictors of the SAR in a hall using multivariate logistic regression. We included median household size, proportion of students in medical courses, hall size and the proportion of students sharing a bathroom with one or more students as covariates.

We used binomial logistic regression on the binary data indicating the presence of at least one infection in each household to estimate the probability that infection is reported by household size. We estimated the binomial probability of secondary infections in a household. We also considered multivariate logistic regression performed with covariates of household size, time between start of term and date of first reported test in the household, and proportion in the household sharing a bathroom. We aggregated household data across halls and only included reports that were associated with symptomatic SARS-CoV-2 infection, to avoid bias in time between start of term and date of first reported test in the household from asymptomatic testing programmes.

We repeated each multivariate regression while at least one predictor was not significant, dropping the predictor with the lowest $t$-value. We performed the statistical analyses using the general purpose mathematical programming language Matlab [10] (logistical analysis) or statistical data analysis software Genstat [11] (binary logistical analysis).

### 2.2.3. Results

#### 2.2.3.1. Reported confirmed attack rate by hall

While all covariates listed in table 7 were significant in a univariate analysis (appendix B), only hall size and proportion of students sharing a bathroom were associated with SAR in the final multivariate regression (table 1). We provide the predicted impact of hall capacity and the proportion sharing bathrooms in table 2. This indicated that students in halls where they all share a bathroom with at least one other (shared = 100%) are approximately 50% more likely to become infected than students in halls with all en-suite rooms (shared = 0%). Increasing the hall capacity from 100 to 400 students increased each student's probability of becoming infected by approximately 167%.

Our results—with the caveat that they are subject to any bias in confirming and reporting infection—suggest that infection risk in large residential settings is difficult to mitigate by segmenting students into households, and the risk of living in large residential settings is exacerbated by the use of shared bathrooms. It is possible that our covariates are proxies for other properties of the setting that influence student mixing (e.g. other types of shared spaces, ventilation, etc.). Furthermore, it is likely that effect sizes will vary between settings depending on importation of cases, characteristics of the

**Table 1.** Coefficients, and associated *p*-value and standard error for the final logistic regression models for the hall SAR.

| covariate | coefficient | *p*-value | s.e. |
|---|---|---|---|
| hall size | 0.0037 | <0.0001 | 0.00006 |
| proportion shared bathroom | 0.4738 | <0.0001 | 0.1166 |
| constant | −3.1466 | <0.0001 | 0.2235 |

**Table 2.** Expected impact of increasing hall capacity (size) and proportion of students sharing a bathroom (shared) on the hall SAR (95% CI) from the final multivariate logistic regression in table 1.

| size | 0% | 50% | 100% |
|---|---|---|---|
| 100 | 0.06 (0.04–0.08) | 0.07 (0.06–0.09) | 0.09 (0.07–0.11) |
| 200 | 0.08 (0.07–0.10) | 0.10 (0.09–0.12) | 0.13 (0.11–0.14) |
| 400 | 0.16 (0.13–0.19) | 0.19 (0.17–0.22) | 0.23 (0.20–0.27) |

local epidemic and local testing facilities, and propensity to adhere to guidance on isolation and mixing restrictions. However, interpreted at face value, our results suggest that only partially filling student residential halls could significantly reduce transmission risk, especially if this is coordinated to reduce shared spaces.

### 2.2.3.2. Infection risk within hall households

Unsurprisingly, the probability of at least one reported symptomatic infection in a household was significantly correlated with household size (table 3); the expected probability of importation into a household of size 16 was 0.68, approximately double the probability for a household of size 8. Thirty-eight per cent of households reported at least one infection.

Household size does not reach significance in the regression model for household SAR in the univariate or multivariate analysis, consistent with estimates of community household SAR for households from population level data [12]. For the multivariate regression we find that SAR was higher for households with the first reported case earlier in the term (table 3). This has many possible drivers such as changes in local background prevalence, shifts in contact or reporting behaviour, or the impact of local depletion of susceptible individuals owing to immunity or students vacating term-time residences. Our analysis of this dataset does not allow us to distinguish between these possibilities. Multivariate regression also indicated the SAR was positively correlated with the proportion of shared bathrooms in the household. The first reported infection in a hall household occurred six days after the start of term. At this stage of the term our predicted household SAR is 0.09 (95% CI: 0.05–0.16) and 0.21 (95% CI: 0.14–0.30) in households with all en-suite rooms and all rooms with shared bathrooms, respectively.

Although the vast majority of test results within a household were dated within 14 days of the first reported positive, and therefore plausibly epidemiologically linked, we did not have any contact tracing or situational data that could be used to investigate this. We have not estimated overdispersion in the number of secondary household cases which may be relevant [13]. While our estimates of the SAR early in the term are broadly consistent with community household SAR (e.g. [12,14]), the binomial probability of reporting a symptomatic infection given a previously reported symptomatic infection in a household over the entire term is lower: 0.058 (95% CI: 0.043–0.070) or 0.076 (95% CI: 0.064–0.090) considering all reported positive tests. However our data on secondary household infections is incomplete due to missing data on household membership and uncertain propensity to report test results (including any time and household dependence of this). Follow-up testing of household members for markers of historic infection in serum samples is probably required to estimate the full extent of household transmission.

It is highly plausible that not all infections in a household arise from a single imported case. In appendix B, we consider the role of infection within the hall on the household SAR using a simple transmission model that allows for infectious contact between household members and between hall

**Table 3.** Coefficients, and associated *p*-value and standard error for final regression models for the probability of introduction of SARS-CoV-2 into a household and household SAR.

| covariate | coefficient | *p*-value | s.e. |
| --- | --- | --- | --- |
| binary logistic regression: probability of infection in household | | | |
| household size | 0.1623 | <0.001 | 0.0269 |
| constant | 1.847 | <0.001 | 0.2510 |
| logistic regression: household SAR | | | |
| date of first infection | −0.1485 | <0.0001 | 0.0298 |
| proportion shared bathroom | 0.9500 | 0.0021 | 0.3091 |
| constant | −1.4028 | 0.0019 | 0.4524 |

members. Results indicate the extent extra-household contacts in the hall may inflate estimates of the SAR; in this model the mean probability of infection due to random contact within the hall is 0.047, whereas the probability of infection from an individual in the same household is 0.091 (see appendix B). In reality, students will also mix with students in other residential settings and with the wider community—we explore evidence for the latter in §2.3.

## 2.3. Transmission to/from the community: comparison with local age groups

Following a series of large outbreaks among the university student population in the 2020/2021 academic year, a question of interest to both policymakers and the general public was the extent to which these outbreaks affected the wider local communities. This question remains of importance for any future large-scale returns of students to their campuses, and provides insight into the extent to which cluster outbreaks impact nearby populations.

In this section, we examine *spillover*, the impact of outbreaks in student populations on the surrounding communities, by analysing patterns of cases among the student population and the local community. In practice, as student populations are interlinked with the wider community, transmission can be in either direction. In addition to any NPIs in place, and adherence thereto, the existence and strength of any spillover signal will probably depend on factors such as: the magnitude of the student outbreak, the levels of newly reported cases (incidence) in the community at the time of the outbreak, and the proportion of students who originally resided in close geographical proximity to the university.

### 2.3.1. Data and methods

We used age-stratified positive case data at the lower tier local authority (LTLA) level from a Public Health England (PHE) line list to describe the trends in student-aged case numbers. Our analysis also used cumulative incidence data as reported by the respective universities, or via the University and College Union (UCU) COVID-19 dashboard [8]. Cumulative case counts from both data sources were used as measures of the outbreak sizes. Calculations of these sizes were limited to 10 days past the peak in student-aged cases in order to facilitate comparisons across all LTLAs.

The age-stratified line list data for those aged 18–24 was used as a proxy for 'student cases', with cases among all other age groups being classified as 'community cases'. To facilitate comparison across age groups, we rescaled all quantities by the known populations of each LTLA using data from ONS [15].

We include a sample of LTLAs with a notable proportion of students in table 4 as an illustration of the variability across England. For each LTLA, we examined if, following an outbreak in the student population, (a) there was an appreciable increase in the growth rate of community cases, and (b) if more community cases than expected were recorded in the subsequent 10 days.

The time-varying growth rate in cases was estimated by taking the derivative of a smoother applied to the daily case data. This method, while accounting for overdispersion in the data, also estimated a mean daily incidence (see appendix C for more details).

Upon infection, a host triggers progeny infections following a period termed the generation time. Changes to the community growth rate (a) were regarded as temporally linked with a student outbreak if such significant changes occurred within two generation times (approx. 10 days [17]). Cases in excess of the expected daily incidence were used as a proxy for (b).

**Table 4.** Properties of each of the considered LTLAs. Local students refers to those students domiciled in the same English region, as obtained from the Higher Education Statistics Agency. The community prevalence was obtained at the regional level from the ONS [16], looking at the transition from 15 September 2020 to 15 October 2020. Multiple return dates arise from those LTLAs which host multiple universities.

| LTLA | region of England | local students | ONS prevalence (%) | return dates |
|------|-------------------|----------------|--------------------|--------------|
| Birmingham | West Midlands | 52.8% | $0.08 \rightarrow 0.79$ | 21 Sep |
| Bristol | South West | 23.3% | $0.08 \rightarrow 0.30$ | 21 Sep & 5 Oct |
| Durham | North East | 15.4% | $0.34 \rightarrow 1.24$ | 5 Oct |
| Exeter | South West | 32.0% | $0.08 \rightarrow 0.30$ | 14 Sep |
| Leeds | Yorkshire & The Humber | 38.7% | $0.25 \rightarrow 1.51$ | 28 Sep |
| Manchester | North West | 50.0% | $0.44 \rightarrow 1.83$ | 14 Sep & 21 Sep |
| Newcastle | North East | 45.7% | $0.34 \rightarrow 1.24$ | 28 Sep |
| Nottingham | East Midlands | 32.1% | $0.13 \rightarrow 0.69$ | 21 Sep |
| Oxford | South East | 37.1% | $0.09 \rightarrow 0.43$ | 5 Oct |
| Salford | North West | 76.6% | $0.44 \rightarrow 1.83$ | 14 Sep |
| Sheffield | Yorkshire & The Humber | 39.3% | $0.25 \rightarrow 1.51$ | 28 Sep |
| York | Yorkshire & The Humber | 33.3% | $0.25 \rightarrow 1.51$ | 28 Sep |

### 2.3.2. Results

The degree to which the growth rate of community cases changed following a student-aged outbreak varied significantly across the studied LTLAs. A selection of the different observed patterns are included in appendix C.

Figure 3 shows a diverse pattern of spillover, and lack thereof, across different English LTLAs. Unsurprisingly, some of the universities with the largest outbreaks were situated in LTLAs which simultaneously had higher levels of incidence in the community.

Larger outbreaks correlate with a greater degree of spillover, although this effect is more strongly seen when considering cases among 18–24-year-olds in figure 3b compared with using reported student outbreak sizes in figure 3a. However, there are exceptions to this pattern, and there is not a clear formal relationship between spillover and outbreak size.

Although we consider two separate data sources to gauge campus outbreaks (self-reported or age-stratified), the discussion below uses outbreak sizes from figure 3a. At lower levels of community incidence, we observe two scenarios: in the first, a small outbreak with little apparent impact on the community. In the second, an outbreak in excess of 1200 cases with the largest observed impact on the community. In this latter case, the impact was larger in relative terms, but not necessarily in absolute terms (net increase in community cases).

No clear relationship is apparent between the proportion of local students and excess community cases.

Some large outbreaks (in excess of 1750) took place with relatively low levels of excess community cases. It is hypothesized that the asymptomatic testing strategy in place at the university in question may have played a role in this outcome.

### 2.3.3. Limitations

Student populations are interlinked with the wider community, whereby transmission can occur in either direction. For a given outbreak then, purely from case data it may not be possible to determine whether or not a student population caused or exacerbated an outbreak in the community. Our findings on spillover here are therefore limited to correlations between the growth in positive cases among the student-aged population and the community.

Particular care should be taken when interpreting the relative timings of increased growth rates as done in appendix C, as community cases rose in England during the autumn. In general, our results are limited by the available data, the sample of studied LTLAs, and our chosen indicators of spillover. While the chosen age groups represent those most likely to be students (ages 18–24) and members of

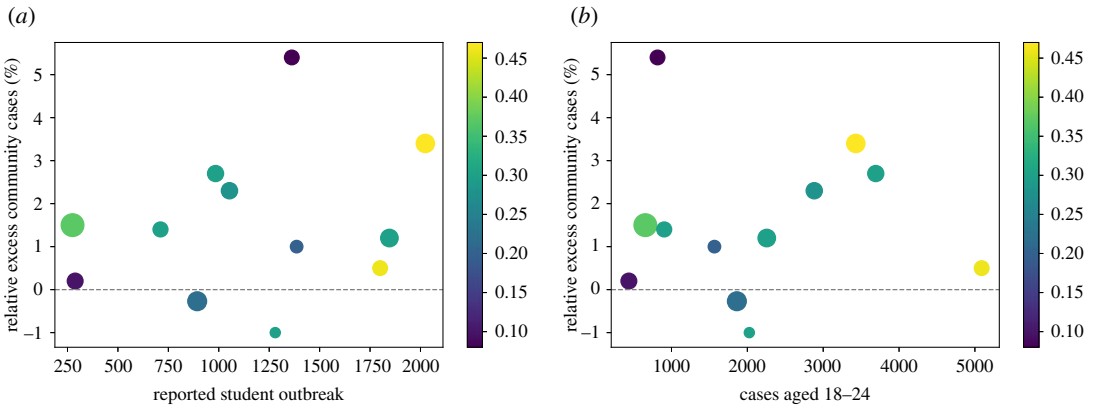

**Figure 3.** Relative excess of community cases in relation to the reported outbreak sizes across the LTLAs considered in table 4. The sizes of the plot markers scale with the proportion of students attending a university in the same region as their home address. The colours of the markers correspond to the community incidence per 1000 people in each LTLA at the time of peak student-aged cases. These inform the varying levels of community prevalence prior to any student outbreak potentially impacting the community. (a) Cumulative university student outbreak sizes up to 10 days past peak incidence, reported by the UCU. (b) Cumulative university-aged outbreak sizes from 14 days prior to 10 days past peak incidence, reported by PHE.

the wider community (ages 0–17 and 25+), these age ranges fail to account for older students and those aged 18–24 who are not in higher education (HE).

Since the analysis is based on confirmed cases, our findings are predicated on consistent testing availability and uptake. Significant changes to these over the studied time period may have impacted our conclusions. The values in figure 3 should not be taken as predictive of the impact a student outbreak will have on the wider community. Overall, signals of spillover are not consistent in type (growth rate or excess) or strength across the studied LTLAs. As such, there does not from the data appear to be a simple set of criteria which can be established to determine the risk to the community from a university outbreak.

While our observations suggest that spillover of cases from the university-aged population to the wider community probably does occur, this analysis does not consider transmission settings, e.g. residential, social or educational.

## 2.4. Transmission to/from the community: spatial patterns

To complement the previous section's spillover analysis based on age-bands, we investigated relationships between the number of cases in areas (middle super output areas, or MSOAs, which are statistical reporting regions in England and Wales typically containing 5000–10 000 people) with a large concentration of students, and areas that are near or far from those student areas.

### 2.4.1. Data and methods

To estimate the proportion of the population within any given MSOA composed of HE students, we used information on the number of people reporting being students in each MSOA from the 2011 UK census [18], and 2019 mid-year population estimates from the Office for National Statistics [15]. For weekly new case counts by MSOA, we used the public UK government coronavirus data portal [19]. We derived MSOA centroids from the Office for National Statistics geographical data [20].

We defined an MSOA as *high student concentration* if the number of students reported in 2011 was at least 15% of the 2019 population estimate, and *low student concentration* if this figure was below 5%. We classified an MSOA as *near* a high student concentration MSOA if it was not itself a high student concentration MSOA but its centroid was within 2 km of the centroid of such an MSOA, and *far* otherwise. We plotted time series of test-positive cases per population by week for these categories of MSOA in several local authorities.

### 2.4.2. Results

We find a very mixed picture across different local authorities hosting HE providers across England, and show several examples in figure 4. In particular, we see some signal of spillover in the case of Manchester

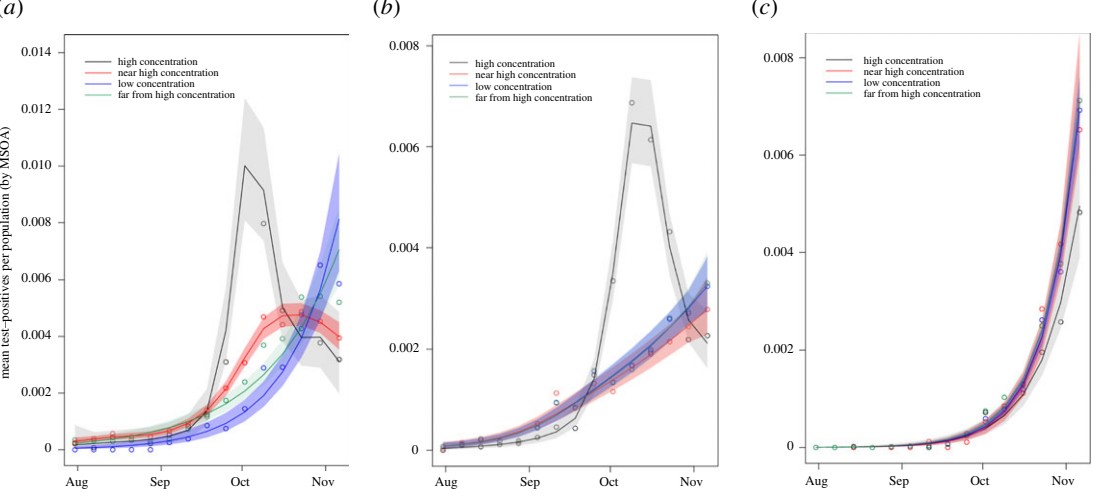

**Figure 4.** Mean cases (represented as dots) per population in MSOAs categorized as high student concentration (black), near high student concentration (red), low student concentration (blue), and far from high student concentration (green) in each of (*a*) Manchester, (*b*) Birmingham and (*c*) Hull. Lines represent the smoothed weekly mean positive cases per population, shaded to cover the 95% confidence intervals of these estimates (details in appendix C).

(figure 4*a*), where the MSOAs near high-concentration student areas experienced a rise and peak in cases following a rise and peak in high-concentration student areas that is visibly distinct from the pattern for areas that are far from student areas. By contrast, in Birmingham (figure 4*b*) we see a rise and peak in cases in high-concentration student areas, but no distinction between the visible patterns for MSOAs near high-concentration student areas or those further away. In the case of Hull (figure 4*c*), we see no obvious distinction between any of the categories of MSOA. When we combine our age-stratified analyses and these geographical-spread analyses we continue to see a mixed picture: some local authorities have signal of spillover, but some do not. We do not see a consistent pattern across England, probably due to wide variations both in the course of the coronavirus pandemic and the nature of university–community interaction in different local authorities. Considerations such as the severity of imposed NPIs, magnitude of student body, and uptake and efficacy of testing, tracing and quarantining measures probably all influence the overall results, but their individual contributions are not identifiable in this analysis. There is agreement between the age-stratified and geographical-spread analyses of spillover in e.g. Manchester and Birmingham. This supports the robustness of the spillover signals (where observed), and the utility of both methods.

## 3. Exploratory modelling for future return

During the autumn term in the 2020/2021 academic year, one of the recurring problems that universities encountered was the large number of students that needed to isolate in halls of residence. The isolation was seen as detrimental to the mental health of students, but also the sheer number of isolated students posed logistical problems to the universities. For instance, making sure that students received adequate food packages was a problem at the beginning of the term. It was an ongoing discussion how to reduce the number of students in isolation and to 'flatten' spikes in the number of isolated students to help universities to better deal with these logistical challenges.

The large outbreaks in universities during the first term in the 2020/2021 academic year led to consideration of methods to safely manage the return of students for the second term of the academic year in January 2021. Two constituent components of the initial guidance (published on 2 December 2020) were the staggered return of students and increased usage of rapid tests [21]. Universities were asked to stagger their students returning over a five-week period according to course type. Students in subjects that most required face-to-face interactions, such as medical and veterinary students, were identified to be the first ones to return to campuses. Guidance also stipulated that all students should be offered a SARS-CoV-2 test when they returned to university, helping identify and isolate those who were asymptomatic. The protocol involved two lateral flow tests (LFTs), 3 days apart. In practice, however, this staggered return did not occur as planned in January 2021. Following the imposition of a new

nationwide lockdown on 4 January 2021, there was a prioritization of return of students to face-to-face teaching enrolled on courses that were most important to be delivered in-person in order to support the pipeline of future key workers. All other courses were to continue being delivered online [22].

In this section, we bring together insights from multiple independent models assessing the impact of staggering the return of students to university and mass testing on infection and isolation. The intention of our modelling work was to focus purely on unpicking the epidemiological consequences of staggering student return on SARS-CoV-2 transmission and isolation. We acknowledge there are multiple factors that administrators must consider and there may be operational and/or resource reasons why a staggered return at higher education institutions is desired. These include ensuring that testing capacity is sufficient to meet demand, the monetary costs associated with the intervention (e.g. testing and staffing) and the educational needs of the students. Though the inclusion of these considerations is beyond the scope of our study, they are important constituents of a multi-faceted decision-making process and we provide an expanded discussion in the Conclusions section.

We present work from four independent models that implement a staggered student return, with the view of having multiple approaches (with distinct modelling assumptions) to enhance result robustness and to determine whether consensus findings emerged. We open with two parsimonious model frameworks. The first is used to highlight potential surges in the number of students in isolation upon student return (§3.1). The second presents a transmission model that considers the impact of staggered student return over time (§3.2). The final two models continue the exploration of the dependency of epidemiological outcomes on staggered return policies, with both models incorporating heterogeneity in contact structure and being partly parametrized using data on (different) individual higher education institutions (§3.3).

With respect to mass testing, we consider insights from two network transmission models, each with a differing area of focus. In one analysis we vary the return testing strategy, in conjunction with staggered student return (§3.4). The other considers regular rounds of testing throughout the academic term and the potential implications of a SARS-CoV-2 variant with increased transmissibility, in light of the emergence of the B.1.1.7 SARS-CoV-2 lineage that proliferated rapidly in the UK in late 2020 and early 2021 [23–26] (§3.5).

## 3.1. Impact of staggering on isolation

To investigate the viability of a staggered return approach, we built a basic discrete event simulation for the return of students to their halls of residence. This individual-based model was designed to investigate the necessary capacity that would be required on campus to isolate incoming students and to establish whether staggering could reduce the overall time that individuals would spend in isolation upon return. In this section, we purely focus upon isolation as a result of a positive test upon return and do not consider spread of infection within the university after students return.

### 3.1.1. Methods

In the model, each student arrives in their household and is tested immediately. If their test is positive, their household is put into isolation for 10 days. If a particular student is due to arrive in a household that is already isolating, that student is required to wait until the relevant household comes out of isolation before they are allowed to return and have their test.

We investigated four different scenarios: (i) all students return on the same day, (ii) each student returns on a random day in a 14-day interval, (iii) each student returns on a random day in a 28-day interval, and (iv) the students return in three weekend 'pulses'. In these pulses, we assume that 10% of students are in halls already and 40% arrive on the first weekend. The next 30% arrive on the weekend three weeks later and the final 20% arrive on the weekend after that. For the purposes of testing, we treat students that are already in halls the same as the first arrival group. In all cases, we assume that the students that come back at a certain point in time are uniformly distributed over the different households. So, we do not consider effects that appear when, for instance, student housing is organized by programme or year. We note that a fully random distribution of returns over a longer period might be practically infeasible, and assuming that returns are concentrated on, for instance, weekends, is a more plausible assumption.

We simulated these scenarios for cohorts of 1000 students. We varied the household size and the probability of receiving a positive test. The results of these simulations are summarized in table 5,

**Table 5.** Summary of the staggering simulations. The table shows the average over 100 runs for each combination of household size and fraction of positive tests (3WP: three week pulsed return, $p$: probability of positive test result, $W + I$: combined total for either waiting to return or isolating).

| household size | $p$ | arrival | isolating | waiting | $W + I$ | peak isolating |
|---|---|---|---|---|---|---|
| 10 | 0.01 | 3WP | 621 | 99 | 720 | 102 |
| | | at start | 931 | 0 | 931 | 170 |
| | | random14 | 594 | 218 | 812 | 94 |
| | | random28 | 577 | 129 | 706 | 89 |
| | 0.02 | 3WP | 1183 | 186 | 1369 | 170 |
| | | at start | 1812 | 0 | 1812 | 320 |
| | | random14 | 1152 | 401 | 1553 | 178 |
| | | random28 | 1171 | 255 | 1426 | 116 |
| | 0.05 | 3WP | 2908 | 438 | 3346 | 303 |
| | | at start | 4049 | 0 | 4049 | 510 |
| | | random14 | 2868 | 951 | 3819 | 307 |
| | | random28 | 2793 | 595 | 3387 | 256 |
| 20 | 0.01 | 3WP | 1190 | 187 | 1378 | 184 |
| | | at start | 1806 | 0 | 1806 | 320 |
| | | random14 | 1151 | 435 | 1586 | 183 |
| | | random28 | 1048 | 250 | 1299 | 167 |
| | 0.02 | 3WP | 2328 | 383 | 2712 | 279 |
| | | at start | 3224 | 0 | 3224 | 500 |
| | | random14 | 2275 | 815 | 3090 | 302 |
| | | random28 | 2103 | 494 | 2597 | 228 |
| | 0.05 | 3WP | 5512 | 875 | 6387 | 520 |
| | | at start | 6352 | 0 | 6352 | 780 |
| | | random14 | 5408 | 1757 | 7165 | 552 |
| | | random28 | 5198 | 1214 | 6412 | 512 |

where we give the total number of days that students need to spend in isolation, need to wait before arriving in their term-time accommodation, and the peak number of students that were in isolation.

We note that, from an organizational perspective for student accommodation, not only the total days spent in isolation is relevant, but also the number of students that are isolated at any given time.

### 3.1.2. Results

To show the impact we have plotted the average numbers for the different simulations in figures 5–7 for the random return within 14 days, 28 days and the three-pulse return.

We observed that staggering the return of students can have organizational advantages. Under a regime where the fraction of positive tests in the student population is low and household sizes are small (figures 5–7, top left panels), spreading out the return of students can reduce the total number of days that students spend in isolation and also reduce the peak number of students that are isolated on a given day. These advantages diminish or are even reversed if the proportion of positive tests is high (figures 5–7, bottom rows); in that case households are repeatedly put in isolation, which leads to higher peaks and total days in isolation. As can be seen in the case of household sizes of 20 students and positive test probability of 0.05, spreading the return of students over a longer period of time mainly reduces the peak number of isolations and does not contribute significantly to a reduction in the total number of days that students are isolated in these scenarios. We note that for positive test probabilities of $p = 0.02$ and $p = 0.05$, one can expect that a significant number of students

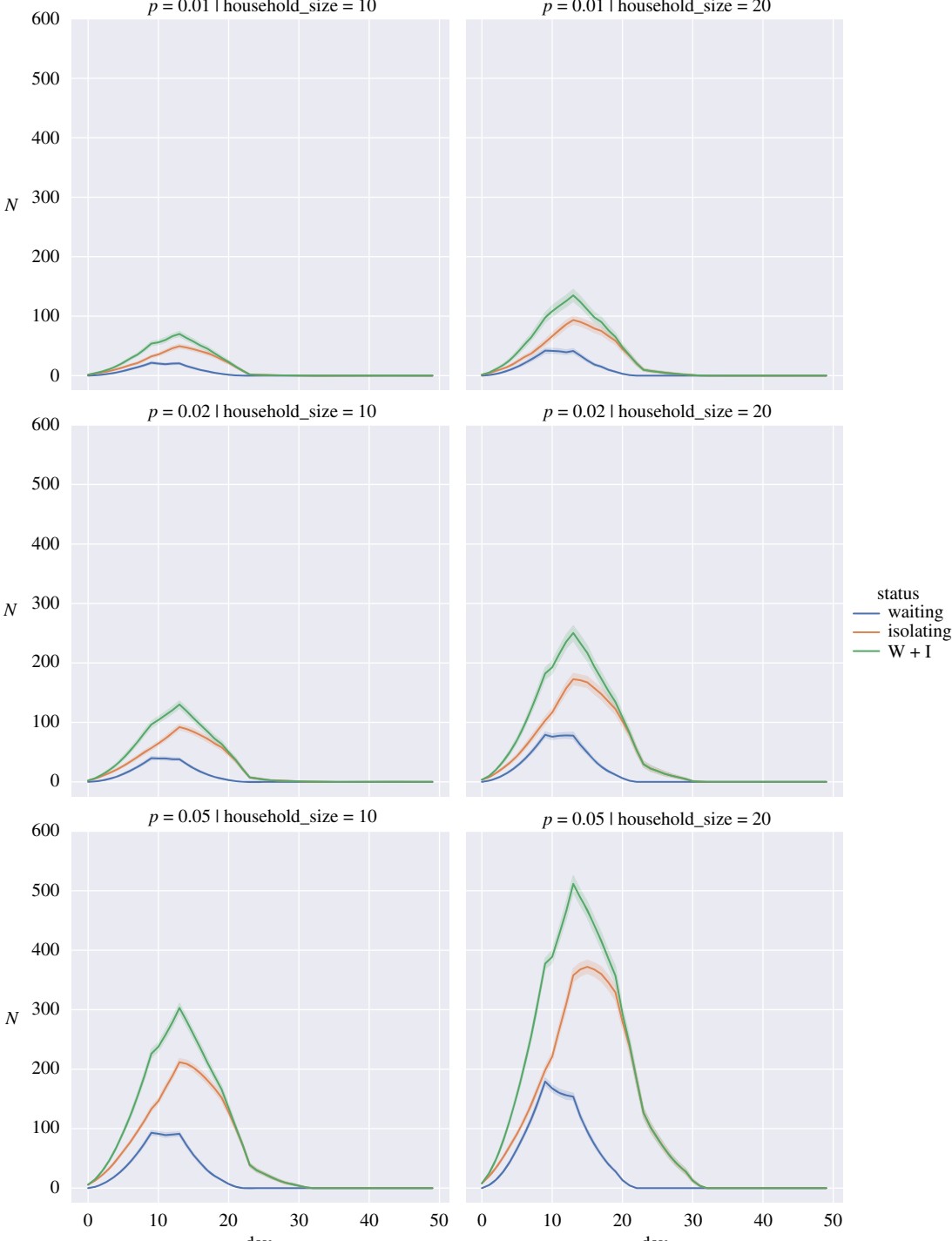

**Figure 5.** Expected number of students in isolation against time for a return spread over 14 days when the probability of a returning student being infected, *p*, is 0.01 (top row), 0.02 (middle row) and 0.05 (bottom row) for household sizes of 10 individuals (left column) and 20 individuals (right column). Waiting (blue), Isolating (orange), W + I: Waiting + Isolating (green). Bands show 95% interval computed from 100 simulation runs.

will be impacted by isolation measures in the first weeks after return. Hence, these results suggest that it is important to take this lead time into account when planning in-person teaching activities.

## 3.2. A simple model for the impact of a staggered student return on incidence

We provide an analysis of the impact of a staggered return of students in three stages, on the transmission dynamics during an academic term.

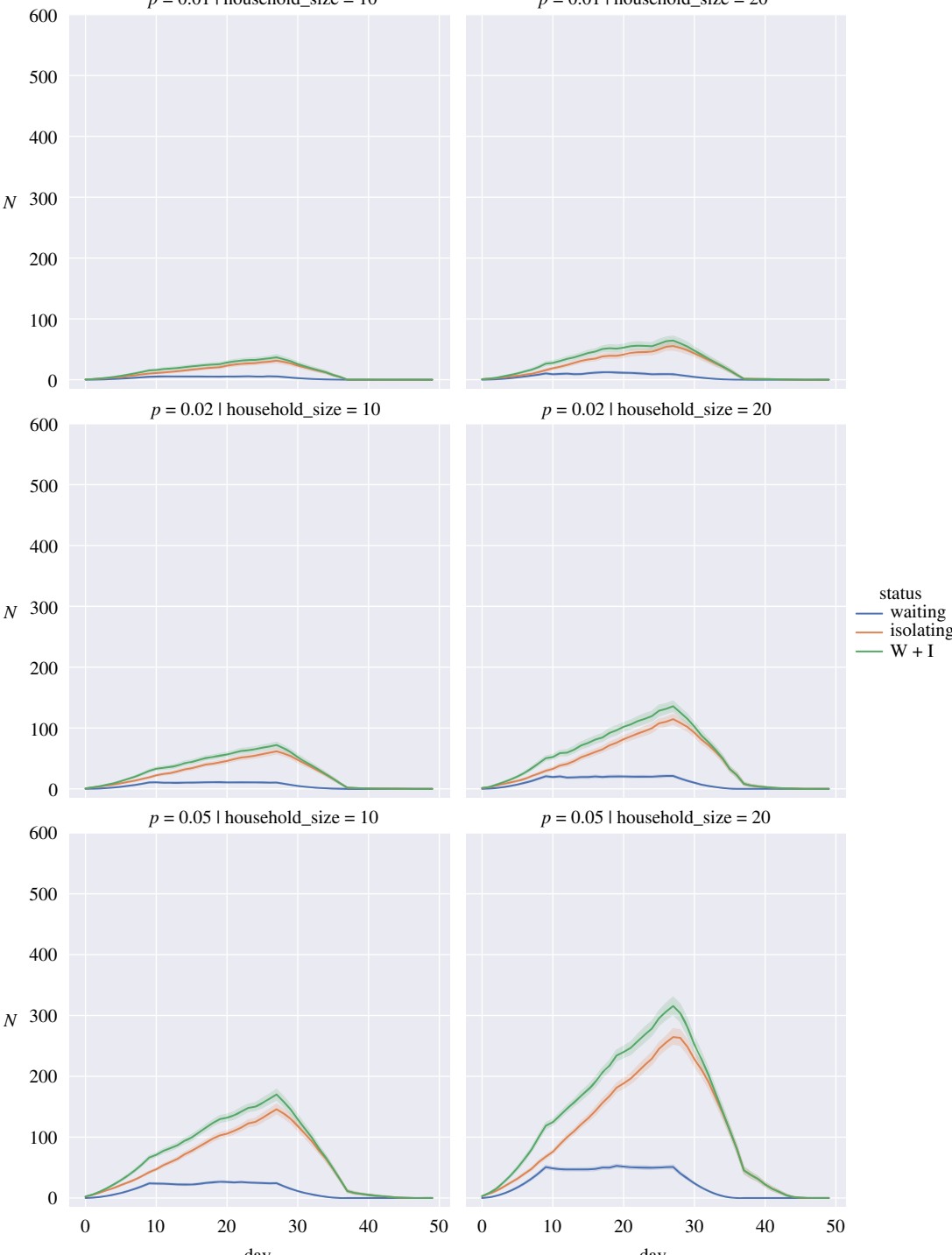

**Figure 6.** Expected number of students in isolation against time for a return spread over 28 days when the probability of a returning student being infected, *p*, is 0.01 (top row), 0.02 (middle row) and 0.05 (bottom row) for household sizes of 10 individuals (left column) and 20 individuals (right column). Waiting (blue), Isolating (orange), W + I: Waiting + Isolating (green). Bands show 95% interval computed from 100 simulation runs.

### 3.2.1. Methods

We implement staggered return of students in a simple compartmental transmission model that segments hosts into susceptible (*S*), infectious (*I*) and recovered (*R*) classes, and examine the mean field solutions of this SIR model. We assume that the students return to university in three stages over three weeks. On return, they mix freely with the existing student body and with each other. At each return point, we assume that a fixed proportion of the returnees are infected.

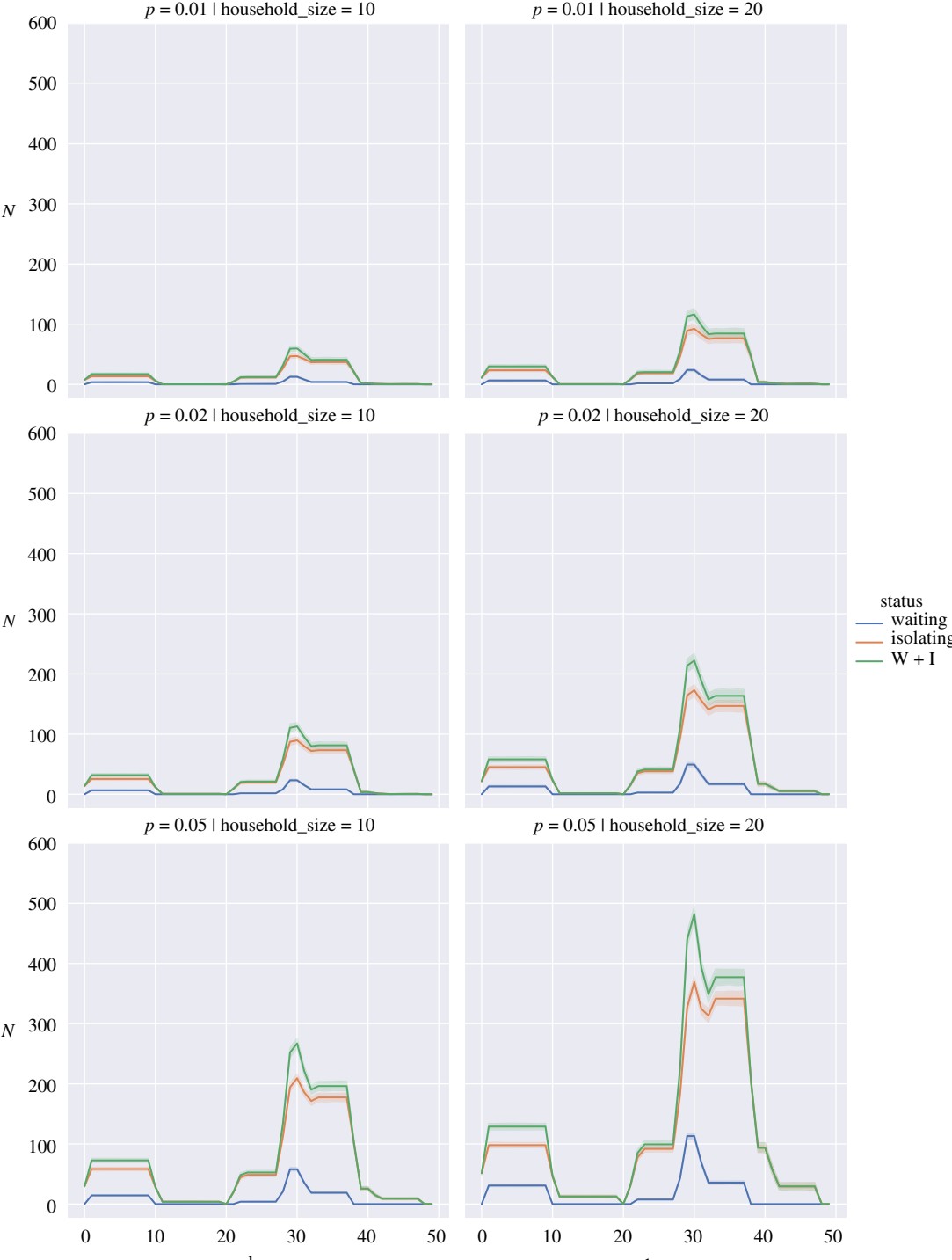

**Figure 7.** Expected isolations for a three week pulse return when the probability of a returning student being infected, *p*, is 0.01 (top row), 0.02 (middle row) and 0.05 (bottom row) for household sizes of 10 individuals (left column) and 20 individuals (right column). Waiting (blue), Isolating (orange), W + I: Waiting + Isolating (green). Bands show 95% interval computed from 100 simulation runs.

In our simulation, we took a student body of $N$ students. These returned in groups of $N/3$ in weeks one, two and three, so that the respective student populations in the first three weeks were $N/3$, $2N/3$ and $N$. Once all of the students return they remain at university for a further eight weeks until the end of an 11-week term.

At each return point, we assume that a fixed proportion, $p$, of the returnees were infected. In full, when each group of $N/3$ students returned they were assumed to contribute $p$ $N/3$ students to the

number of infected students, and $(1 - p)N/3$ students to the susceptibles. The resulting SIR model is then given by the following three-level piece-wise model, where $t = [0 \ldots 77]$ was measured in days, $i = 1$ gave the infection dynamics in week 1, $i = 2$ the infection dynamics in week 2, and $i = 3$ the infection dynamics in weeks 3–11:

$$\frac{dS_i}{dt} = -\beta \frac{IS}{N_i},$$

$$\frac{dI_i}{dt} = \beta \frac{IS}{N_i} - \gamma I$$

and

$$\frac{dR_i}{dt} = \gamma I.$$

In each of the three stages, the population values were: $N_1 = N/3$, $N_2 = 2N/3$ and $N_3 = N$.

To simulate the staggered returns, we took the values of $S$ and $I$ at the start of the first, second and third weeks to be the following, noting that the values of $S$ and $I$ then jump at the start of each week (as can be seen in the figures):

$$S_1(0) = (1 - p)N/3, \quad I_1(0) = \frac{pN}{3},$$

$$S_2(7) = S_1(7) + (1 - p)N/3, \quad I_2(7) = I_1(7) + \frac{pN}{3}$$

and

$$S_3(14) = S_2(14) + (1 - p)N/3, \quad I_3(14) = I_2(14) + \frac{pN}{3}.$$

For simulation examples, we used a population size of $N = 1000$ and considered three scenarios with different values of the prevalence ($p$) and transmissibility ($\beta$): (i) $p = 0.10$, $\beta = 0.18$; (ii) $p = 0.02$, $\beta = 0.18$; (iii) $p = 0.02$, $\beta = 0.30$. In all scenarios, we fixed the recovery rate $\gamma = 0.072$. We also compared the results of the 'staggering' model with that of an unstaggered model (with the same parameter values) in which all of the $N$ students returned at the start of term.

### 3.2.2. Results

The corresponding reproduction numbers $R$ for the three scenarios are initially: (i) $R = 2.5$, (ii) $R = 2.45$ and (iii) $R = 4.08$.

In the absence of all other controls, and across all three considered scenarios, we observed that staggering can slightly reduce and slightly delay the size of the infection peak in the short term (figures 8–10). However, over the course of the 11-week term the reductions in the overall attack rate were minor, particularly for infections with high transmissibility (figure 10).

While based on relatively simple assumptions, these results are intuitive. In conclusion (i) a staggered return could delay and reduce the outbreak peak, however, (ii) without other controls, staggering will not much reduce the overall attack rate over the course of an academic term.

## 3.3. Structured models assessing the impact of a staggered student return

The formerly presented parsimonious models provide guiding principles on the potential impact of staggering on infection throughout the course of an academic term and isolation upon return. In this section, we build on the prior work by investigating the role of staggered student return on epidemiological outcomes using models incorporating additional layers of complexity. In contrast to the compartmental model in §3.2, these models are simulated probabilistically to explore the random/ stochastic variation in outcomes. Specifically, we used two models of transmission dynamics for SARS-CoV-2 in a university setting, each using a different model conceptualization: (i) a stochastic compartmental model [6] and (ii) a network-based model [27]. Both transmission models assume that upon exposure hosts enter a period of latent infection during which they are not infectious, then hosts may remain asymptomatic throughout their infection (asymptomatic cases) or transition through presymptomatic and symptomatic stages of infection. Mass asymptomatic testing may detect both presymptomatically infected hosts and asymptomatic cases. Note that both models assumed that individuals did not 'compensate' by replacing contacts that were unable to occur (due to the expected contact being in isolation or not having yet returned to the university setting).

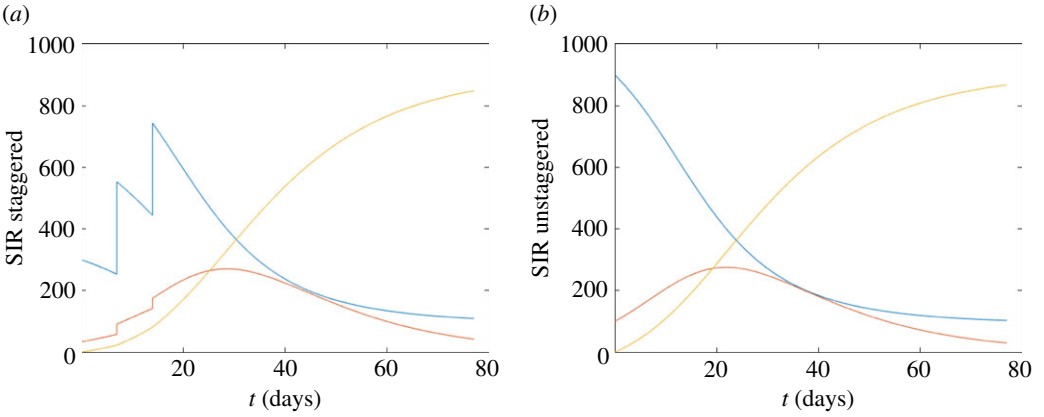

**Figure 8.** Staggered/unstaggered return temporal profiles. (*a*) Staggered return (three stages in the first three weeks) in the first three weeks, taking $N = 1000$ and $\beta = 0.18$, $p = 0.1$, $\gamma = 0.072$, with an initial value of $R = 2.25$. (*b*) Unstaggered return. In each figure, we show S (blue), I (red) and R (yellow).

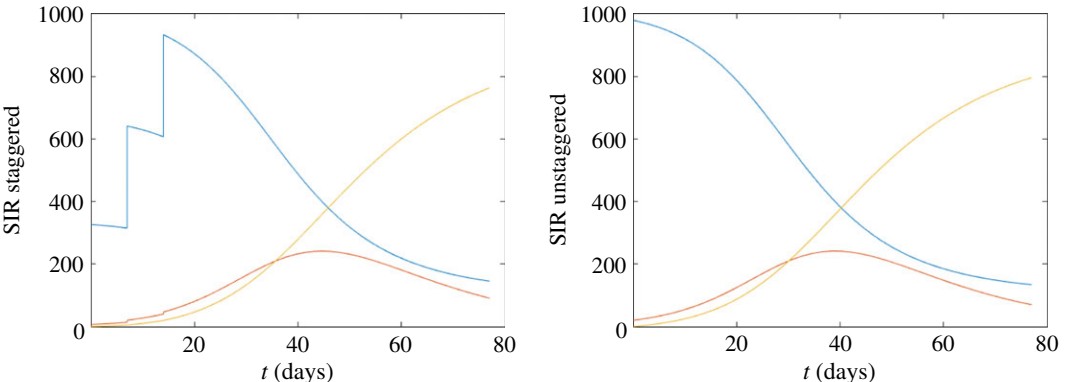

**Figure 9.** Staggered/unstaggered return temporal profiles. $\beta = 0.18$, $p = 0.02$, initial $R = 2.45$.

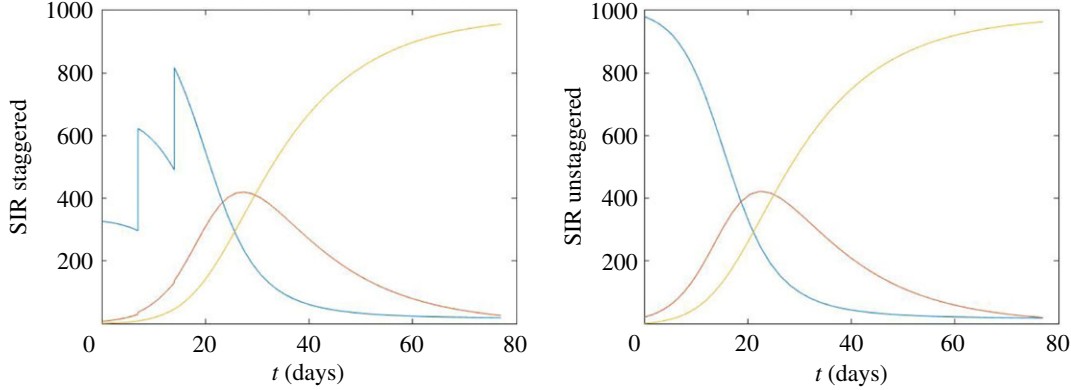

**Figure 10.** Staggered/unstaggered return temporal profiles. $\beta = 0.3$, $p = 0.02$, initial $R = 4.08$.

### 3.3.1. Methods

#### 3.3.1.1. Stochastic compartmental model summary

The stochastic compartmental model included realistic mixing patterns for students based on student responses to the Social Contact Survey conducted in 2010 [28,29]. These contact matrices entailed 160 groups based on school (department) and year of study, with contacts stratified into household, study and random contacts. We calibrated the disease compartments to estimations made at the start of the 2020/2021 academic year in the absence of controls, returning an R of approximately 3 (for calibration we assumed asymptomatic cases were 50% less infectious than symptomatic cases). Further model details, including descriptions of the remaining assumptions underpinning the model, may be found in [6].

For our analysis here, we fixed the mean probability of a case being asymptomatic at 75% and the relative infectiousness of an asymptomatic we varied between 0 and 1. It was assumed that the university would operate within Public Health England guidelines and therefore that symptomatic cases would be tested and self-isolate within 48 h. Students in large halls of residence were assumed to be restricted to households of 24 individuals, reflecting actions taken by universities in the 2020/2021 academic year. We did not include the impact of contact tracing, social distancing or the use of face coverings. We used a student population size of 28 000. The number of infected students at the start of term was estimated using home location and incidence as of July 2020 as described in [6] using an anonymized extract of student data for a specific university relating to the 2019/2020 academic year. The study complied with the University data protection policy for research studies [30]. Each scenario was run for a simulated 300 days, with 10 replicates per scenario.

The model is coded in R and C++ and available at https://github.com/ellen-is/unimodel.

### 3.3.1.2. Network model summary

Our network model framework represents interactions between students within a university population in different settings (household, study cohort, organized societies and sports clubs, other social). We ran an epidemic process on this network, for the virus SARS-CoV-2. The model includes isolation and contact tracing. We adopted a pessimistic approach by assuming a comparable amount of mixing to pre-pandemic circumstances, and did not include any reduction in the risk of transmission occurring over contacts due to social distancing and/or the use of face coverings.

Specifically, we assumed students had contact with all household members each day. We sampled the number of non-household contacts from distributions fit to data informed by student responses to the Social Contact Survey conducted in 2010 [28,29], with stratification according to the level of study (undergraduate or postgraduate). For this analysis, we then applied the following two contact pattern changes to all but the baseline (no intervention) scenario: (i) society contacts did not occur (transmission risk therefore zero), assuming that all meetings would take place online; (ii) for on-campus resident students, we assumed no contacts within the broader accommodation unit of the same floor or block of residence (thus outside the immediate household).

In all simulations, we had an overall student population of 25 000, with 7155 students resident on-campus and the remainder off-campus. Each simulation run had a duration of 11 weeks, encompassing both a 10-week academic term and the week prior to its commencement.

We initialized latent, infectious (asymptomatic, presymptomatic and symptomatic) and recovered individuals using estimates for 2 January 2021 from the University of Warwick SARS-CoV-2 transmission model [31], based on fits from 29 November 2020 and assuming no change to adherence in NPIs.

For each parameter configuration, we ran 1000 simulations, amalgamating 50 batches of 20 replicates; each batch of 20 replicates was obtained using a distinct network realization. We performed the model simulations in Julia v. 1.4–1.5. The data and science surrounding the SARS-CoV-2 infection is fast moving. This piece of sub-analysis was originally undertaken in December 2020, with our intent being for this work to provide a record of the state of our modelling at that time. For a full description of the network model and noted limitations of the methodology, see [27]. We summarize in appendix D other changes made from the base model to carry out this analysis. Distributions of outcome measures are visualized using violin plots which capture the smoothed probability density of a set of numeric values [32].

### 3.3.1.3. Staggered return strategies

We assessed four strategies for the return of students for the academic term (figure 11) using the stochastic compartmental model and the network-based model. Note that, across all considered strategies, a proportion of the student population was considered to be resident in university accommodation between academic terms.

The four strategies were as follows: (i) no stagger—for students not resident in university accommodation over the vacation, they return on day 1; all students entered the return test procedure on day 1 (we acknowledge that in practice there would be logistic difficulties associated with such a strategy); (ii) 14-day spread—each student is allocated their day to return to university (if applicable) and they begin the return testing procedure between days 1 and 14 (sampled according to a uniform distribution); (iii) 28-day spread—similar to the 14-day spread strategy, except the applicable range spans days 1 to 28; and (iv) three-weekend pulse (by course)—fractions of the student population return on designated weekends based on level and course of study. In the stochastic compartmental model, for the

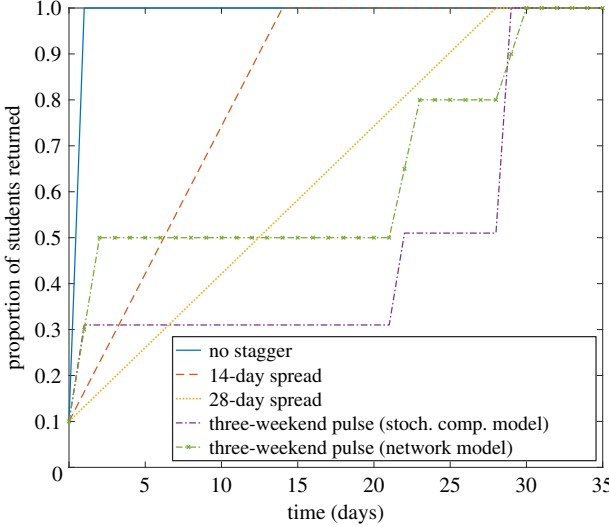

**Figure 11.** Staggered return temporal profiles. We considered four student return patterns: no stagger (blue solid line); return spread over 14 days (orange dashed line); return spread over 28 days (yellow dotted line); three-weekend pulsed return (by course), as used in the stochastic compartmental model (purple dot-dash line); three-weekend pulsed return, as used in the network model (green dot-dash line, cross markers). For this depiction, we present proportion returned with respect to time when assuming 10% of all students were resident in their university accommodation between academic terms.

three-weekend pulse, on day 1 of the simulation we assumed that all vital medical, dental and veterinary students enrolled in courses (as provided by the University of Bristol [33]) were present, as well as 20% of students in all other schools, giving 31% of students present at university in total. This first group of students was chosen because they were studying on the courses that were allowed to return when universities were closed in January 2021 and at this time it was estimated that 20% of students who were not enrolled on these courses still chose to return. On day 22 of the simulation, all other courses with important practical elements return to university, giving a total of 51% of students present at university. On day 29, all remaining students return to university. For the network model, we set the groupings (and the associated proportion of students returned) for the three-weekend pulse as a variation on the University of Warwick plan for staggering student return [34].

### 3.3.1.4. Testing protocols

We also included a testing protocol that adherent students engaged with upon return to university. In the stochastic compartmental model, we considered two scenarios: (i) no testing on student return, and (ii) testing of all non-symptomatics. We assumed the tests detect half of true positives (50% sensitivity) and do not generate false-positive results (100% specificity).

In the network model, we assumed adherent students underwent two LFTs, 3 days apart, with isolation between tests (for details on test sensitivity and specificity, see appendix D). For each strategy for student return, we sampled the proportion of students that were adherent to isolation from zero compliance (value 0) to full compliance (value 1) in increments of 0.1. We assumed an identical adherence to isolation restrictions independent of the cause (presence of symptoms, household member displaying symptoms, or identified as a close contact of an infected by contact tracing). Additionally, we assumed those that would engage with isolation measures would also engage with contact tracing.

### 3.3.2. Results

### 3.3.2.1. Stochastic compartmental model results

We first present our findings from simulations carried out with the stochastic compartmental model. The collection of simulations that we present here give an indication of what the impact of staggering and testing might have been at the start of the 2020/2021 academic year, if this had taken place. The model parameters do not change based on events that have happened since this time, including vacation periods, and consequently the results are to be interpreted qualitatively if used to make predictions about future scenarios.

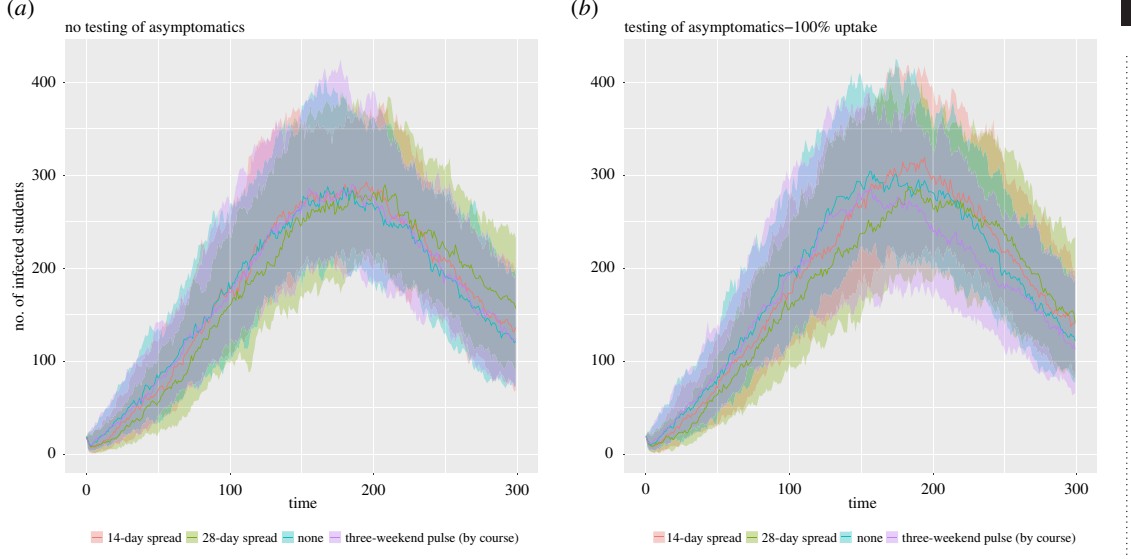

**Figure 12.** Epidemiological outcomes among a student population given differing staggered return strategies to university using a stochastic compartmental model. Outputs are summarized from 10 simulations, with the lines representing the median number of symptomatic and asymptomatic students and the shaded areas showing the 2.5th and 97.5th percentiles. We display distributions corresponding to: (a) no testing of asymptomatics upon student return, (b) all asymptomatics are tested.

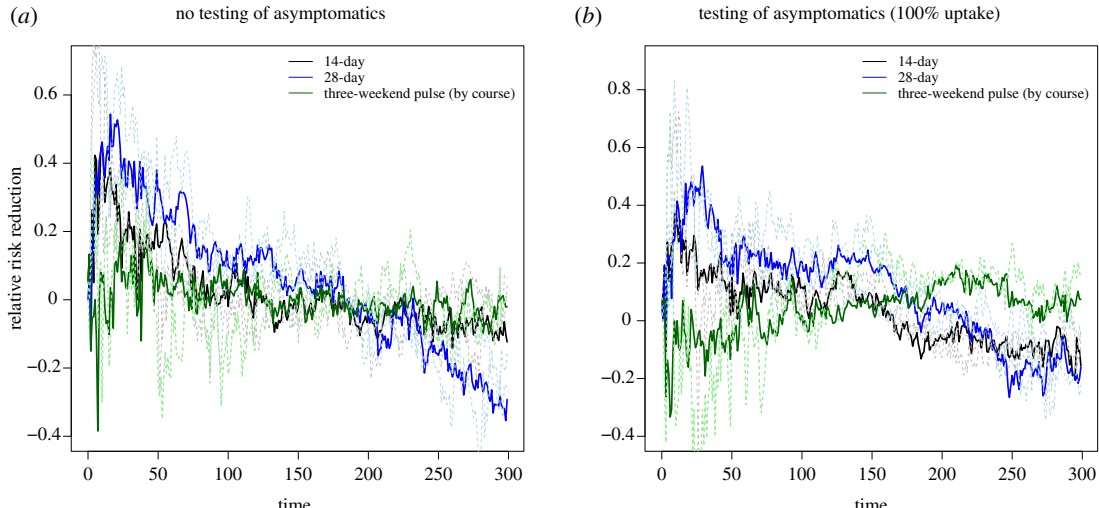

**Figure 13.** Epidemiological outcomes among a student population given differing staggered return strategies to university compared with a strategy where staggering is not used, using a stochastic compartmental model. Outputs are summarized from 10 simulations, with the continuous lines representing the median number of symptomatic and asymptomatic students and the dashed lines corresponding to the 2.5th and 97.5th percentiles. We display distributions corresponding to: (a) no testing of asymptomatics upon student return, (b) all asymptomatics are tested.

We observed a similar overall case burden across all considered staggering strategies. Given high adherence to control, similar temporal trends were observed regardless of the testing strategy used (figure 12). Relative to an unstaggered return, there was lower prevalence in the early phase paired with higher prevalence in the late phase for the 14-day and 28-day strategies, with these relationships being consistent across the collection of test upon student return protocols (figure 13).

### 3.3.2.2. Network model results
For the independent analysis performed using the network model, on account of the inherent uncertainty in several parameters of the model and assumptions made regarding contact patterns, we once more

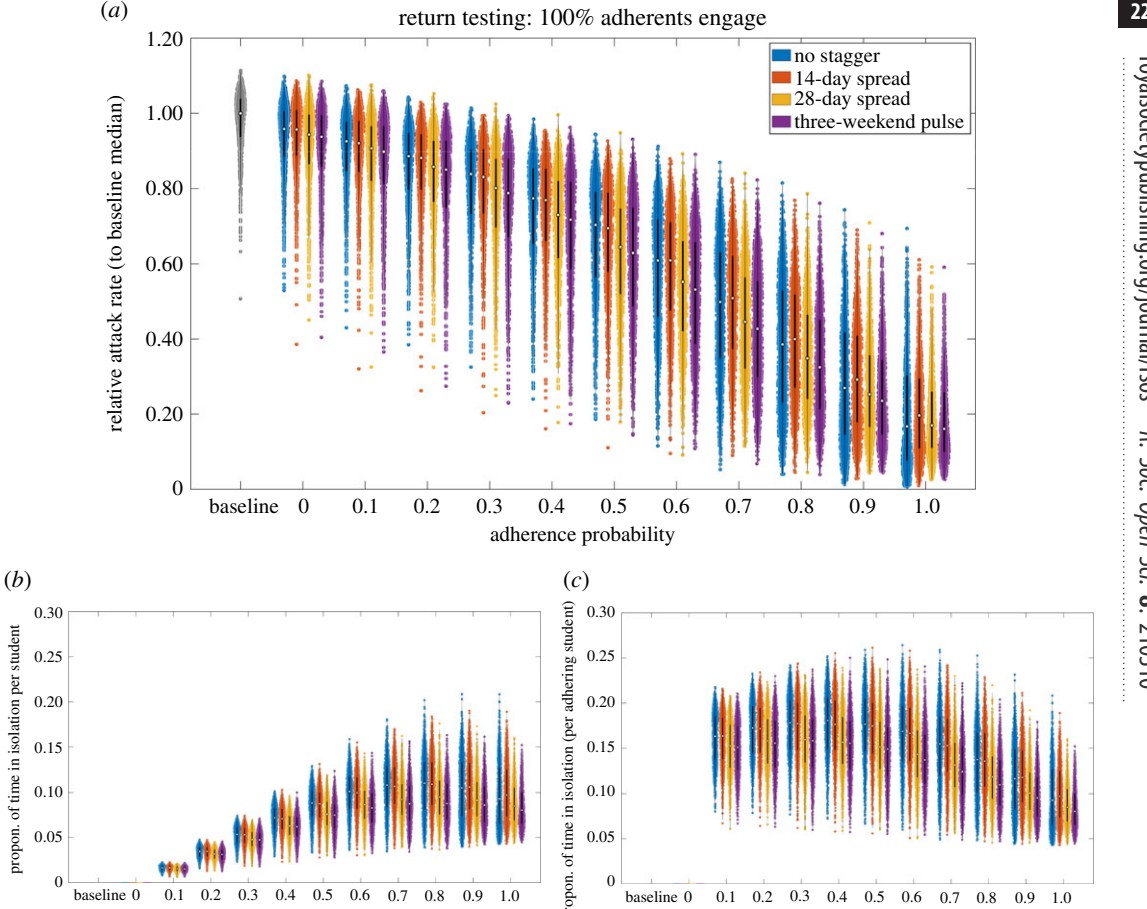

**Figure 14.** Epidemiological outcomes among a student population given differing staggered return strategies to university. Outputs summarized from 1000 simulations (with 20 runs per network, for 50 network realizations) for various levels of adherence to NPIs. We considered four strategies: no stagger (blue violin plots); return spread over 14 days (orange violin plots); return spread over 28 days (yellow violin plots); three-weekend pulsed return (purple violin plots). We assumed 100% of adherents engage with return testing. We display distributions corresponding to: (*a*) relative attack rate, compared with the baseline scenario; (*b*) time spent in isolation per student; (*c*) time spent in isolation per adherent student. The white markers denote medians and solid black lines span the 25th to 75th percentiles.

focus on qualitative comparisons across the simulated scenarios (as done with the stochastic compartmental model). We first note that, compared with the baseline scenario, the scenario with reductions in contacts via organized societies and dynamic on-campus accommodation contacts (represented by adherence probability 0.0 in figure 14) produced a shift downwards in the obtained distributions of relative attack rate (medians of 0.93–0.96 across the four staggering strategies).

Comparing attack rate across staggering strategies for a fixed adherence level, in concordance with the stochastic compartment model we found a minimal impact on the attack rate over the course of the academic term. Furthermore, we determined adherence to isolation guidance and following test and trace procedures as crucial in reducing the overall case burden within the student population (figure 14*a*).

Assessing the potential impact of staggered return strategies on the amount of time students may be required to isolate, for a fixed adherence level there were no substantial differences between the strategies we considered (figure 14*b,c*). Inspecting a measure of time spent in isolation for any given student, we observe an initial increase with adherence level, peaking when roughly 70–80% of students are adherent, before declining as it approaches all students being adherent (figure 14*b*). A collective response (high adherence) reduced the time each adherent student was estimated to spend in isolation, compared with a scenario of moderate adherence among the student population (figure 14*c*).

In the absence of other interventions, staggering slightly reduces and delays the size of the peak, though the long-term impact is minimal (figure 15*a*). For strong adherence to interventions, temporal trends were found to be broadly similar regardless of the staggering strategy used (figure 15*b*), in

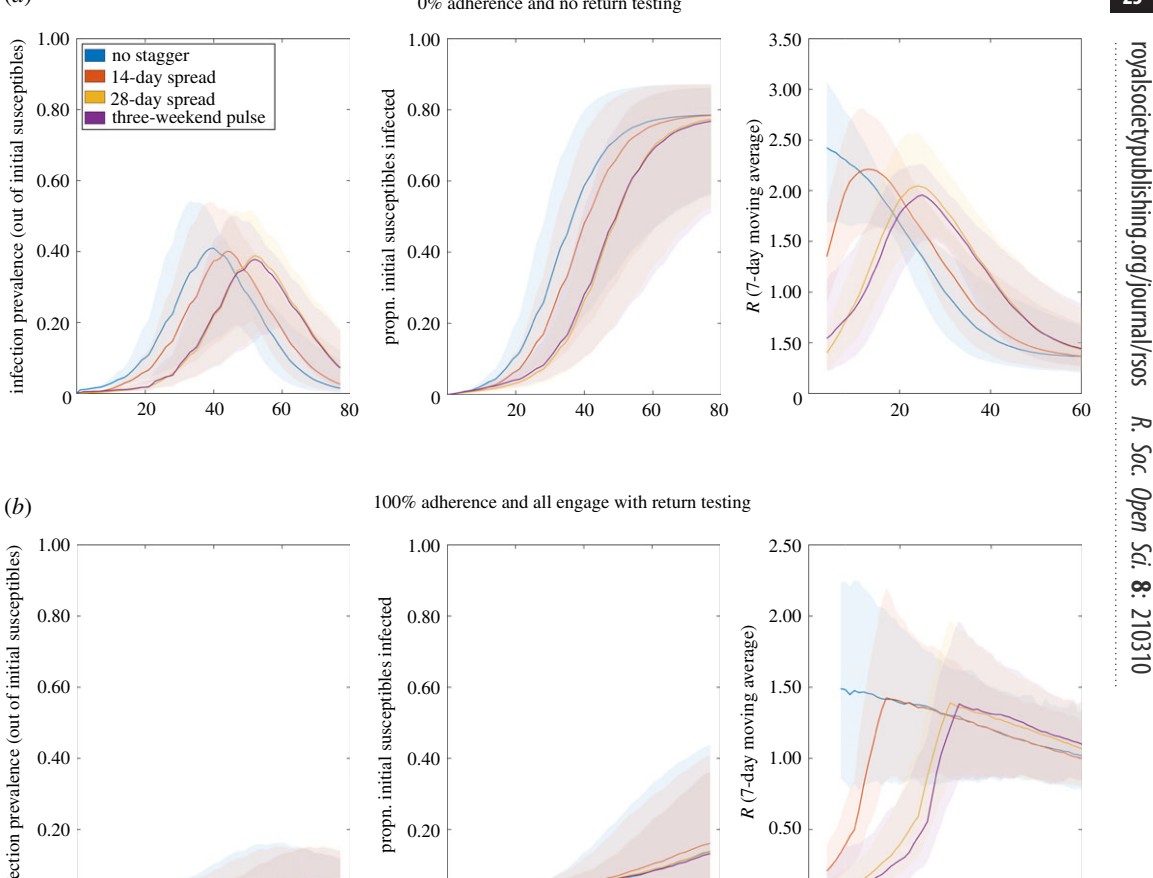

**Figure 15.** Temporal profiles of epidemiological measures over the academic term under differing return patterns. Outputs produced from 1000 simulations (with 20 runs per network, for 50 network realizations) for four return patterns: no stagger (blue); return spread over 14 days (orange); return spread over 28 days (yellow); three-weekend pulsed return (purple). Solid lines depict the median profile and shaded regions the 95% prediction interval. Panels from left to right display infection prevalence, cumulative proportion of initial susceptibles infected, and 7-day averaged $R$, respectively. (a) No return testing; (b) return testing with all adherents participating.

agreement with the temporal trends observed from the stochastic compartmental model projections (figure 12).

## 3.4. Testing on return

Using the network model described in §3.3, we modelled implementation of a testing protocol that students would be advised to complete before attending face-to-face teaching.

### 3.4.1. Methods

To investigate the sensitivity of staggered returns to alternative test-on-return strategies, using a fixed high level of adherence (90%), we investigated four protocols (table 6). Test protocol A: two LFTs, 3 days apart, with isolation between tests (the default assumption); test protocol B: single LFT; test protocol C: two LFTs, 3 days apart, with no isolation between tests; test protocol D: single PCR with isolation until test result received (2-day delay), leaving isolation upon a negative test result.

### 3.4.2. Results

Given high adherence to interventions and engagement with rapid testing, the inclusion of a second LFT and isolation between the LFTs gives minor reductions in attack rate (comparing A–D in figure 16). We

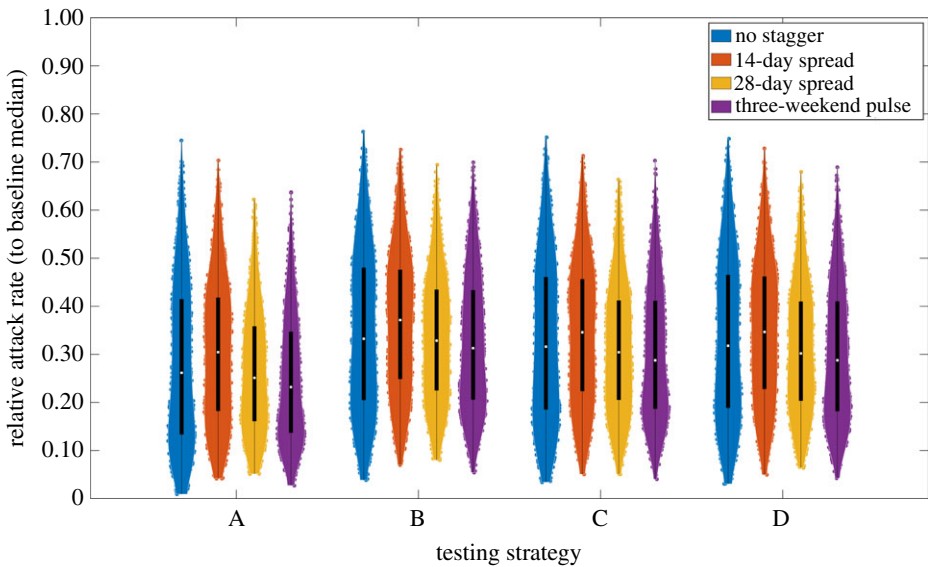

**Figure 16.** Relative attack rate distributions under different test before return to study procedures, in combination with strategies for staggered student return. Assumed 90% adhere to isolation, test and trace guidance. For test strategies using two LFTs, the two tests were spaced 3 days apart. We considered four student return patterns: no stagger (blue violin plots); return spread over 14 days (orange violin plots); return spread over 28 days (yellow violin plots); three-weekend pulsed return ( purple violin plots). The white markers denote medians and solid black lines span the 25th to 75th percentiles.

**Table 6.** Overview of the return test protocols. Cells containing an 'X' denote the element being a part of the return test protocol. LFT 1 and LFT 2 correspond to a first and second LFT respectively. Conditioned on the plan including individuals undergoing two LFTs, 'isolate between tests' reflects whether isolation should occur between the two LFTs.

| testing strategy | LFT 1 | LFT 2 | isolate between tests |
|---|---|---|---|
| A | X | X | X |
| B | X | | |
| C | X | X | |
| D | single PCR test | | |

found comparable attack rate distributions across our four ( previously introduced) staggering strategies for student return to university (comparing between colours in figure 16).

## 3.5. Testing during term

To build on our investigation of testing on arrival, we simulated the impact of an asymptomatic testing system in use throughout the term, assuming the presence of a more transmissible SARS-CoV-2 variant. This scenario was considered in response to the emergence of the B.1.1.7 variant in the UK, which began to become widespread from November 2020.

### 3.5.1. Methods

We used a layered network model of contact between 15 000 simulated students, with one layer of household contacts and one of other-group contacts intended to simulate all out-of-household contact. Individuals could be infected by either household or non-household contacts. Infected individuals progressed through disease states via a stochastic compartmental model including a latent period, various infectious states (presymptomatic, asymptomatic or symptomatic) and recovery resulting in immunity (which we assumed did not wane).

We investigated five during-term asymptomatic testing scenarios, in which individuals were tested at random with probability 1/3, 1/7, 1/10 or 1/14 per day (to simulate testing every 3, 7, 10 or 14 days,

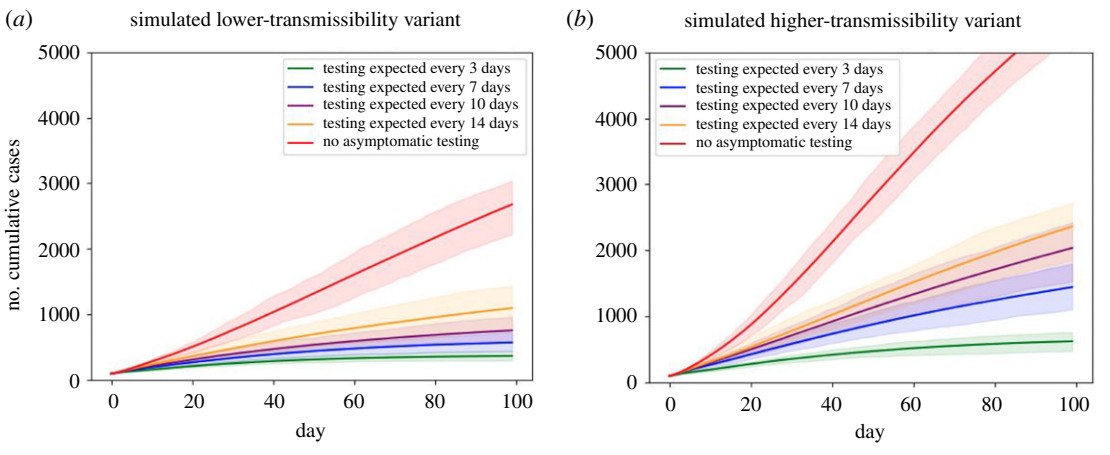

**Figure 17.** Temporal profiles of cumulative case counts for a simulated population of 15 000 students under differing during-term asymptomatic screening scenarios. We present two scenarios for variant transmissibility: (*a*) lower-transmissibility variant; (*b*) higher-transmissibility variant (1.5 times more transmissible than the lower-transmissibility variant). Output produced from 100 runs of each scenario, with a new network generated for each replicate; envelopes show 95% of model runs and solid lines show mean values. Asymptomatic screening scenarios considered are: no asymptomatic testing (red), each person randomly tested with probability 1/14 (yellow), 1/10 (purple), 1/7 (blue), or 1/3 (green) per day, to simulate testing approximately every 14, 10, 7 or 3 days, respectively. *Note that this model has many limitations and should be interpreted mainly qualitatively. See main text for a listing of some limitations.*

respectively), or not at all. In all scenarios, symptomatic individuals are assumed to be tested immediately upon developing symptoms. Upon a positive test, the entire household isolates for 14 days. Simplifying assumptions included perfect and rapid testing and perfect adherence to testing and isolation. We assumed 50% of non-household contacts to be traced and isolated.

We first ran these scenarios with a lower-transmissibility variant intended to plausibly simulate the variant of SARS-CoV-2 circulating in universities in the UK in autumn 2020. We then considered a 1.5 times more transmissible variant, intended to simulate a potentially more transmissible variant such as B.1.1.7.

We initialized each simulation with 100 infectious individuals, and ran the model for 100 timesteps (notionally days). For each scenario, we performed 100 replicates, each run on a newly generated network. Importantly, we chose the particular parameters for this model for a combination of plausibility and simplicity, and some are not well-founded in any particular dataset. Details of the model, parameter choices and limitations are available in appendix E.

### 3.5.2. Results

We plot the number of cumulative cases as a time series under the differing testing scenarios for the two variants in figure 17. In general, more frequent asymptomatic screening better controls cases, with the scenario with no asymptomatic screening seeing the largest number of cases. While cases were contained to a mean of fewer than 1200 in all scenarios with asymptomatic screening in the less transmissible setting, this was only achieved by the most frequent testing scenario in the more transmissible setting.

### 3.5.3. Limitations

This model has many simplifying assumptions and the absolute numbers it produces should not be considered in isolation or as an absolute prediction. Some of these limitations include: perfect adherence to testing and isolation, no vaccination nor prior immunity, no reactive interventions during the course of the simulation, and a speculative network contact structure that *has not been trained from data* but is instead simply a plausible simple structure. In addition, the model did not include a reduction in the risk of transmission occurring over contacts due to face covering use or social distancing; however, other work [35] suggests that if such measures are in place in a university setting

and/or if there are moderate levels of immunity, the impact of testing is less prominent, highlighting the importance of considering testing in the context of other measures.

# 4. Conclusion

The mass migration of students at the beginning and end of academic terms, their unique living arrangements during term time and unique patterns of social mixing, make them an important population for the spread of infectious respiratory illnesses. Despite this, prior to the COVID-19 pandemic, there was little data collected on outbreaks of infectious disease at universities (although one such dataset collected between October 2007 and mid-February 2008 has now been analysed in 2021 [36]) and university students were an understudied population. Therefore, at the start of the pandemic, there was a limited evidence base to support policy decisions around universities. Our study brings together expertise from multiple research groups and presents results from multiple statistical and modelling analyses and provides new understanding on infectious disease outbreaks at universities and how these could be mitigated.

An important finding of our study is that adherence to NPIs is likely to have more impact than staggering the return of students to university. Survey data suggest that in the autumn term of 2020, students generally did have high adherence to NPIs; an Office for National Statistics (ONS) survey found high adherence (90%) to social distancing across multiple universities [37]. In addition, a survey of University of Bristol students found that 99% of students self-isolated after testing positive for COVID-19 and the majority of survey participants reported low contact numbers [38]. However, there was heterogeneity in adherence, with some students reporting many contacts and with only 61% of students with cardinal COVID-19 symptoms self-isolating [38]. In future, it will be important that students maintain their high levels of adherence and to ensure they have sufficient resources to allow them to do so.

Several of the scenarios presented here have considered the frequency of asymptomatic screening at universities. This has been explored in other modelling studies, for example [39] found that monthly screening can reduce cumulative incidence by 59% and weekly screening by 87%. We found that increasing the frequency of asymptomatic screening is likely to be important in the presence of a more transmissible SARS-CoV-2 variant, with cases only being able to be maintained below 1200 (mean cumulative over 100 days) when testing occurs every 3 days (in a population of 25 000). This finding corroborates a study that used an agent-based model to simulate COVID-19 transmission at the University of California San Diego, where larger outbreaks resulted in a maximum outbreak size of 158 when asymptomatic screening occurred monthly and 7 when it occurred twice weekly [40], but with a much lower impact seen on the average outbreak size when increasing from monthly to twice weekly testing, ranging from 1.9 to 1.1, respectively. Brooks Pollock *et al.* [6] also found that mass testing was more effective for higher values of the reproduction number. This highlights the importance of reassessing control measures under different variants.

We have focused here on COVID-19 risks and mitigation strategies for when students return to university and during the university term itself; however, we have covered little on the risk of transmission from infected students to private homes at the end of term. Previous modelling work suggests that in an unvaccinated population, an infectious student would on average generate just less than one secondary within-household infection, but this is dependent on the prevalence in the student population at the time of departure [41]. Although it is expected that vaccination will reduce the impact of students returning to private homes at the end of term, the UK vaccination programme is ongoing, and there are particular spatial areas and demographic groups where low uptake is expected [42], suggesting that this still may be an important question to consider in future.

Our analyses and discussions have highlighted several areas that we recommend for further attention. These include building a better understanding of determinants of adherence, including attributes that may place subpopulations at higher risk (e.g. students in part-time employment). Given the need for rapid turnaround of our analyses, a persistent challenge is the ability to access data in a timely manner and ensuring any barriers to data access have a purpose and are necessary. One mechanism for addressing this data availability issue may be a centralized nationwide student testing data resource, which could serve as a hub for anonymized student testing data that documents institution and attributes such as type of accommodation.

We recognize there are prominent factors that we have not addressed here, as we have focused directly on transmission dynamics, yet should be considered while viewing our results in a broader context. One

important future research direction is to consider the non-COVID impact of intervention measures. The majority of work to date on COVID-19 has focused upon developing intervention policies that seek to minimize the overall number of cases, hospital admissions or deaths. However, it is important to acknowledge that any control policy that may reduce transmission also has an impact in terms of monetary cost, non-COVID health, mental health and well-being. An extension to this work could focus upon assessing the direct monetary cost of intervention policies as well as the logistical and operational constraints associated with such policies [43,44]. For example, to sustain a regular testing regime at universities under financial, logistical, or structural constraints, mathematical modelling suggests that pooling RT-qPCR testing may be a cost-effective method, although this may come with additional caveats resulting from the associated reduction in sensitivity (when cases are not detected) and specificity (when students self-isolate but are not infected) [45]. Additionally, in higher educational settings, it is important to consider any impact on teaching and examination schedules as well as mental health and well-being of students. The models considered here allow for an estimate of the different resources used by the different control strategies. In order to determine an optimal intervention, it is crucial to establish the objective of any control policy, noting that the objective may not be generalizable across all higher education establishments. Once an objective is appropriately defined, any modelling can be specifically tailored to maximize the robustness of any advice offered.

Furthermore, a growing picture is just beginning to emerge on the prevalence of, and risk factors for, 'long COVID' symptoms and health complications following coronavirus (COVID-19) infection. An initial set of early experimental results collected by the ONS indicates around one in five respondents testing positive for COVID-19 exhibit symptoms for a period of five weeks or longer, and around one in 10 respondents testing positive for COVID-19 exhibit symptoms for a period of 12 weeks or more [46,47]. We recognize that the current university closures may have significant impact upon student mental health and well-being—across multiple surveys collecting information on how the COVID-19 pandemic has affected the mental health of students, a consistent outcome was above 50% of respondents expressing that their well-being and mental health had become worse [48]. In addition, we hope that the ongoing vaccine roll-out will provide a level of protection for those most vulnerable to severe outcomes, which in turn may alleviate risks associated with possible student to community spread.

In conclusion, our findings are comprised of three overarching points. Firstly, we observed evidence of spillover transmission between higher education populations and the wider community in some, but not all, settings. Secondly, we would expect reductions in adherence to NPIs (including case and household isolation) to have more impact than any marginal benefits generated from a staggered return of students to university. Thirdly, the emergence of more transmissible new variants results in impaired effectiveness of mass asymptomatic testing. Ultimately, we hope that the work presented here can be used by universities and policymakers to assist in the long-term strategy of ensuring that students can return safely to their studies at universities in the UK. And while we have focused on the national picture in the UK, we also hope our results can offer insights relevant to higher education in other countries.

Data accessibility. This article has no additional data.

Authors' contributions. J.E. conceptualization, methodology, software, formal analysis, investigation, writing—original draft, review and editing, visualization. E.M.H. conceptualization, methodology, software, formal analysis, investigation, writing—original draft, review and editing, visualization. H.B.S. conceptualization, methodology, software, formal analysis, investigation, writing—original draft, review and editing, visualization. K.J.B. conceptualization, methodology, investigation, data curation, writing—original draft, review and editing, visualization, supervision. E.J.N. conceptualization, methodology, software, formal analysis, investigation, writing—original draft, review and editing, visualization. E.L.F. conceptualization, methodology, software, formal analysis, investigation, data curation, writing—original draft, review and editing, visualization. M.L.T. conceptualization, investigation, writing—original draft, review and editing. E.B.-P. conceptualization, software, review and editing, supervision. L.D. conceptualization, review and editing, supervision. C.J.B. conceptualization, methodology, software, formal analysis, investigation, writing—original draft, review and editing, visualization. R.B.H. conceptualization, methodology, software, formal analysis, investigation, writing—original draft, review and editing, visualization. L.S. conceptualization, methodology, software, formal analysis, investigation, writing—original draft, review and editing, visualization. J.R.G. conceptualization, writing—original draft, review and editing, supervision, project management. M.J.T. conceptualization, methodology, formal analysis, investigation, writing—original draft, review and editing, supervision, project management.

Competing interests. We declare we have no competing interests.

Funding. This work was supported by EPSRC grant no EP/R014604/1. The authors would also like to thank the Virtual Forum for Knowledge Exchange in Mathematical Sciences (V-KEMS) for the support during the workshop *Unlocking higher education Spaces – What Might Mathematics Tell Us?* where work on this paper was undertaken. K.J.B.

acknowledges support from a University of Nottingham Anne McLaren Fellowship. E.L.F. acknowledges support via K.J.B.'s fellowship and the Nottingham BBSRC Doctoral Training Partnership. M.L.T. was supported by the UK Engineering and Physical Sciences Research Council (grant no. EP/N509620/1). E.B.-P., E.J.N., L.D., J.R.G. and M.J.T. were supported by UKRI through the JUNIPER modelling consortium (grant no. MR/V038613/1). E.M.H., L.D. and M.J.T. were supported by the Medical Research Council through the COVID-19 Rapid Response Rolling Call (grant no. MR/V009761/1). H.B.S. is funded by the Wellcome Trust and the Royal Society (grant no. 202562/Z/16/Z). J.E. is partially funded by the UK Engineering and Physical Sciences Research Council (grant no. EP/T004878/1).

Acknowledgements. The authors thank the Isaac Newton Institute for Mathematical Sciences, Cambridge, for support during the programme *Infectious Dynamics of Pandemics* where work on this paper was undertaken.

# Appendix A. A simple outbreak model for university COVID-19 outbreaks

We postulate in §2.1 that the probability, $\mathcal{P}$, that a university experiences a SARS-CoV-2 outbreak is given by

$$\mathcal{P} = 1 - p^n, \tag{A 1}$$

where $n$ is the number of imported cases and $p$ is the probability that an imported case fails to seed an outbreak.

We tested this hypothesis by using estimates for the number of imported student cases [7] and COVID-19 case number data for a number of universities for which cumulative case number data was available on the UCU COVID dashboard [8].

For a university $i$ with cumulative case number $c_i$, we defined an outbreak if $c_i > T_u$, where $T_u$ is a threshold number of cases. We set $x_i = 1$ if a university had experienced an outbreak, and $x_i = 0$ if not.

The probability mass function for the distribution of outbreaks is given by

$$f(x|p) = p^{n(1-x)}(1 - p^n)^x.$$

Thus, the likelihood of the data for all $N$ universities, given $p$, is

$$L(p) = \prod_{i=1}^{N} p^{n_i(1-x_i)}(1 - p^{n_i})^{x_i}$$

and the log likelihood is

$$LL(p) = \sum_{i=1}^{N} \{n_i(1 - x_i) \log p + x_i \log (1 - p^{n_i})\}.$$

Maximizing the log likelihood gives the maximum-likelihood estimate $\hat{p}$ for $p$. The $100(1 - \alpha)\%$ confidence interval for $p$ is given by

$$\left[\hat{p} \pm z(\alpha/2) \frac{1}{\sqrt{NI(\hat{p})}}\right],$$

where $z(\alpha/2)$ is defined by $P(Z > z(\alpha/2)) = \alpha/2$ for $Z \sim N(0, 1)$ and

$$I(p) = -E\left(\frac{\mathrm{d}^2}{\mathrm{d}\,p^2} LL(p)\right) \tag{A 2}$$

$$= E\left(\sum_{i=1}^{N} \frac{n_i}{p^2(1 - p^{n_i})^2} \{(1 - p^{n_i})^2 - x_i(1 - p^{n_i}) + x_i n_i p^{n_i}\}\right) \tag{A 3}$$

$$= \sum_{i=1}^{N} \frac{n_i^2 p^{n_i}}{p^2(1 - p^{n_i})}, \tag{A 4}$$

using $E(x_i) = 1 - p^{n_i}$.

**Table 7.** Coefficients, and associated *p*-value and standard error, for the univariate and intermediate multivariate logistic regression models for hall SAR.

| covariate | coefficient | *p*-value | s.e. |
|---|---|---|---|
| univariate logistic regression: SAR | | | |
| hall size | 0.0037 | <0.0001 | 0.00006 |
| constant | −2.8388 | 0.0001 | 0.1722 |
| median household size | −0.0539 | 0.0029 | 0.0181 |
| constant | −1.3218 | <0.0001 | 0.01590 |
| proportion shared bathroom | 0.3541 | 0.0017 | 0.1097 |
| constant | −1.9836 | <0.0001 | 0.0822 |
| proportion medical faculty | 0.3511 | 0.0004 | 1.7973 |
| constant | −2.5257 | <0.0001 | 0.2238 |
| multivariate logistic regression: SAR | | | |
| hall size | 0.0030 | <0.0001 | 0.0007 |
| median household size | 0.0300 | 0.2458 | 0.0258 |
| proportion shared bathroom | 0.4141 | 0.0010 | 0.1253 |
| proportion medical faculty | 4.0628 | 0.0712 | 2.2521 |
| constant | −3.6647 | <0.0001 | 0.4690 |
| hall size | 0.0030 | <0.0001 | 0.0007 |
| proportion shared bathroom | 0.3977 | 0.0013 | 0.1233 |
| proportion medical faculty | 2.7342 | 0.1588 | 1.9402 |
| constant | −3.2354 | <0.0001 | 0.2795 |

# Appendix B. Additional analyses for student hall infection data

## B.1. Additional regression results

Univariate and intermediate multivariate regression results for the household and hall SAR (§2.2) are summarized in tables 7 and 8.

## B.2. Stochastic transmission model for hall and household infection

An alternative method for exploring the role of household and hall size, discussed briefly in §2.2, is to fit a stochastic transmission model that allows for infection between hall members in addition to household members.

### B.2.1. Methods

We first calculated the household size distribution for each hall. We ignore the temporal dynamics (setting the infectious period to unity) and simulated the final size of the outbreak using the Sellke construction [49] in a population with two levels of mixing, defined by the household infectious contact rate, $\lambda_H$, and the global (or hall) infectious contact rate, $\lambda_G$. Motivated by the lack of dependence of household SAR on household size (table 1), we assumed density-dependent mixing in households. Contacts at each level were assumed to be made at the points of a homogeneous Poisson process. We calculated the probability of a student being infected given a single introduction in the hall. Inference was performed for each hall using the approximate Bayesian computation tutorial in Kypraios *et al.* [50], assuming Exp(1) priors for $\lambda_H$ and $\lambda_G$.

### B.2.2. Results

In figure 18*a*, we plot the probability that a student was infected by another within their hall including their household ($p_{\text{hall}} = 1 - e^{-\lambda_G AR}$) (where $AR$ is the hall attack rate, in this case the number of reported

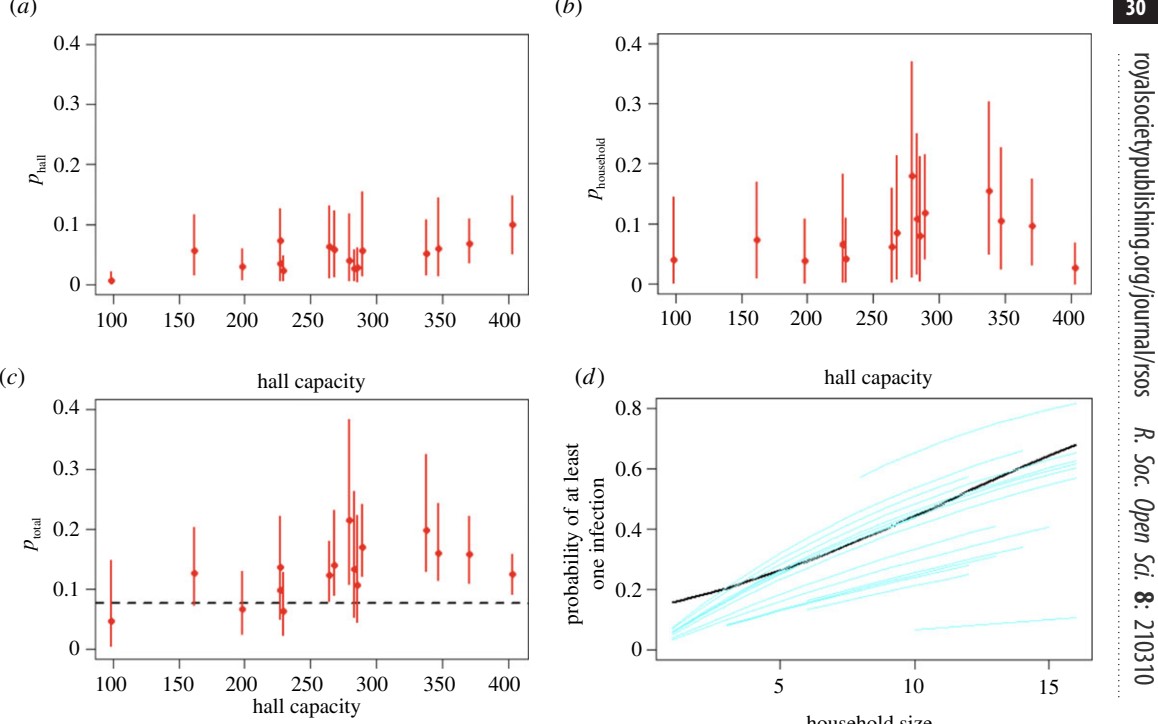

**Figure 18.** Results of fitting the model with two levels of mixing to each hall individually, plotted against hall capacity. Circles indicate expected mean and lines 95% confidence intervals. (*a*) Probability of infection due to global infectious contact. (*b*) Probability of infection due to household infectious contact. (*c*) Comparison of the total probability of infection after introduction accounting for household and global infectious contacts compared with the estimated binomial probability of infection given introduction into a household (black dashed line). (*d*) Comparison of the probability of infection in a household by household size for each hall (blue lines) and the output from the binary regression analysis (black line).

**Table 8.** Coefficients, and associated *p*-value and standard error, for the univariate and intermediate multivariate logistic regression models for household SAR.

| covariate | coefficient | *p*-value | s.e. |
|---|---|---|---|
| univariate logistic regression: SAR | | | |
| household size | −0.0543 | 0.1442 | 0.0372 |
| constant | −2.2885 | <0.0001 | 0.3679 |
| date of first infection | −0.1354 | <0.0001 | 0.0291 |
| constant | −0.8816 | 0.0272 | 0.3996 |
| proportion shared bathroom | 0.7472 | 0.0151 | 0.3074 |
| constant | −3.3660 | <0.0001 | 0.2730 |
| multivariate logistic regression: SAR | | | |
| household size | −0.0743 | 0.0558 | 0.0388 |
| date of first infection | −0.1547 | <0.0001 | 0.0305 |
| proportion shared bathroom | 0.9911 | 0.0015 | 0.3115 |
| constant | −0.6350 | 0.2966 | 0.6084 |

confirmed infections with known household). The additional probability of infection from within their household is shown as $p_{\text{household}} = 1 - \mathrm{e}^{-\lambda_H}$ in figure 18*b*. In figure 18*c*, we plot the probability of infection ($p_{\text{total}}$) accounting for both household and global infectious contacts within an infected household. This is compared with the binomial probability of reporting an infection given a

previously reported infection in a household (which does not distinguish between halls). Figure 18$d$ compares the estimated probability of a household reporting an infection for each hall to the estimation from the binomial logistic analysis (table 3). There is some indication that global infectious contacts may play a relatively greater role in overall infection risk in the largest halls. However, choices for distributing missing household data, which is ignored for here, will probably influence the relative size of $p_{hall}$ and $p_{household}$, as will choices about scaling of mixing intensity with household size.

The maximum hall size in this data is approximately 400 students and findings may not generalize to other hall settings or future periods of student return. Other limitations of this approach are the lack of differentiation between symptomatic and asymptomatic infections, pre-existing immunity, or the impact of isolation, so that parameters are interpreted as averages across students in a hall in addition to the caveats arising from the missing data. Furthermore, we assume a single introduction and a closed system of fixed occupancy, so that any imported cases are attributed to infection within the hall. Dedicated household-based studies in student residential halls would be valuable for untangling the role of mixing within households, halls and with the community on infection risk in these settings.

# Appendix C. Additional information on age-stratified observations

## C.1. Additional observations: age-stratified analysis

### C.1.1. Methodological details

The numerical interpolation method, and the subsequent calculation of the growth rate of positive cases, is applied to the positive case counts in each LTLA, rescaled by the number of people (falling within the considered age range) estimated to live there. This quantity, $c(t)$, shows a consistent day-of-week effect due to e.g. varying test availability and test seeking behaviour. To account for overdispersion in the data, we assume a quasi-Poisson distribution in the fitting.

A smoother $\varrho(t)$ is applied using thin-plate splines, such that $c(t) \propto e^{\varrho(t)+\omega_i}$, where $\omega_i \, \forall \, i \in [1, 7]$ is used to apply a fixed effect for each day of the week. The instantaneous growth rate of the cases is simply given by $\varrho'(t)$. This was implemented using a general additive model from the R package $mgcv$ with a canonical link [51]. Past examples of this method can be found in [52] (see figure 19).

### C.1.2. Growth rates

Despite the clear spikes in cases among 18–24-year-olds across all LTLAs in figure 20, the growth rates for community cases are qualitatively very different. In figure 20$c$, the community growth rate mirrors the national trend. In figure 20$a$, the growth rate of community cases is higher, and appears to lag after the growth in student-aged cases.

In figure 20$d$, a qualitatively different scenario emerges, with a marked rise in the growth of community cases following an outbreak among the student-aged population. Finally, in figure 20$b$, the outbreak among 18–24-year-olds has no perceptible impact on the growth rate of community cases.

### C.1.3. Limitations

The estimated growth rates of confirmed cases, and the estimated excess community cases following a large student-aged outbreak, are sensitive to the choice of the spline in the smoother. Changing the spline does not qualitatively alter our conclusions.

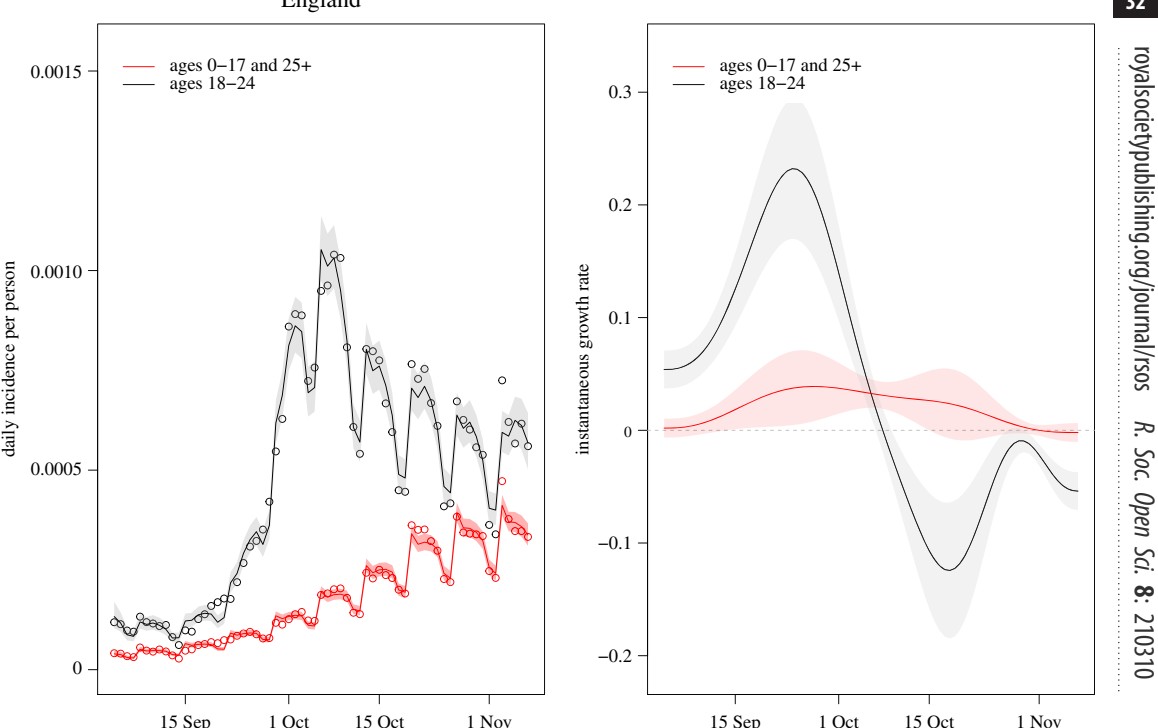

**Figure 19.** Growth rate among the student-aged (grey) and community populations (red) across England. University outbreaks are observed on the national scale, with the higher incidence per population among those aged 18–24. The community growth rate in cases increased from late September 2020. However, there was not a statistically significant subsequent increase following the peak in student-aged outbreaks. The shaded regions are the 95% confidence intervals for the relevant quantity.

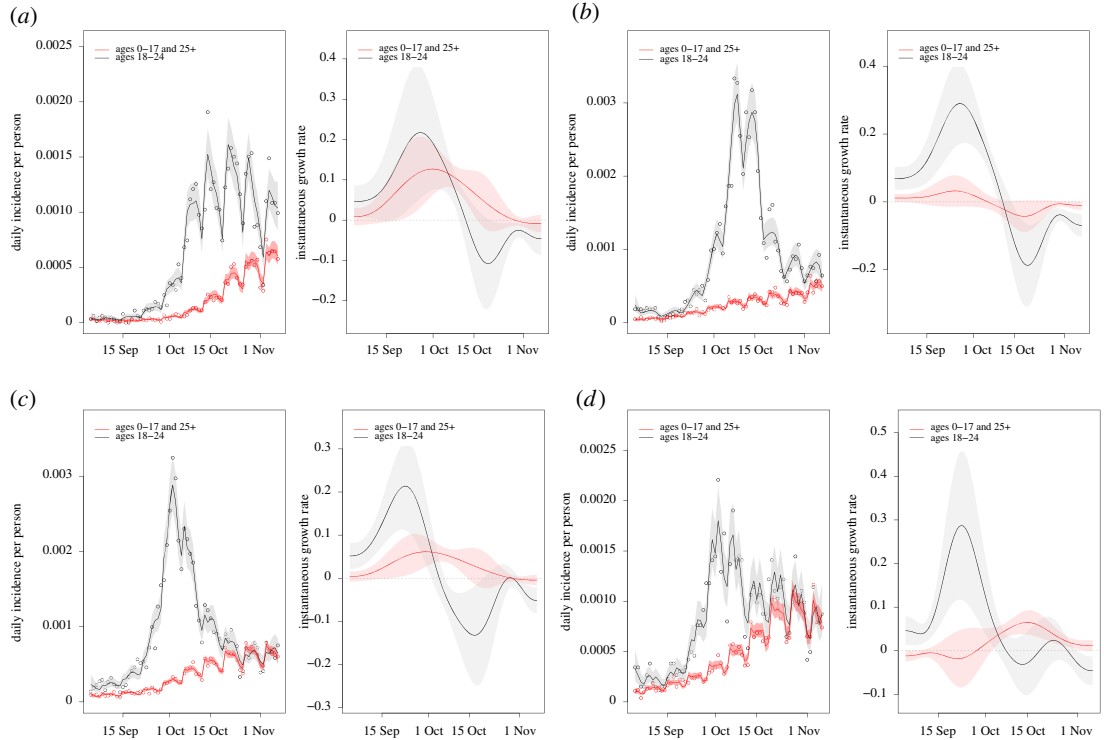

**Figure 20.** Examples of the different types of growth rate patterns observed among student-aged (grey) and community cases (red). The shaded regions are the 95% confidence intervals for the relevant quantity. (*a*) Bristol, (*b*) Durham, (*c*) Leeds and (*d*) Salford.

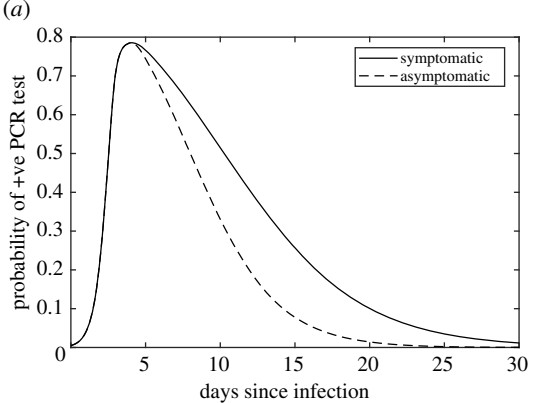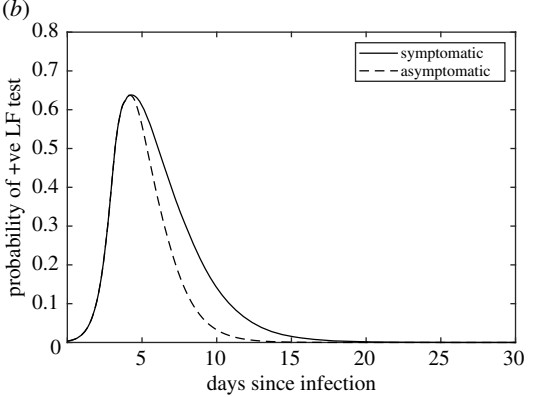

(a)

(b)

**Figure 21.** Probabilities of testing positive through time for symptomatic and asymptomatic individuals. We assumed that the probability of positive test results being returned in symptomatic and asymptomatic individuals were equal during the proliferation stage of the virus, but that the probability of asymptomatic individuals testing positive decayed faster in the clearance stage, owing to a shorter mean clearance duration of 6.7 days [53] (a) PCR test; (b) LFT.

# Appendix D. Additional information on the network-based structured model

## D.1. Test sensitivity

The probability of testing positive is probably a function of viral load; while symptomatic and asymptomatic individuals have similar average peak viral loads and proliferation stage durations, their average duration of clearance stages has been observed to differ [53,54]. Therefore, we used distinct test sensitivity profiles for symptomatic and asymptomatic cases (figure 21). However, we highlight that this is an area of considerable uncertainty. Future studies detailing the testing probability of asymptomatic individuals, and the specific relationship between viral load and testing probability, would be a valuable contribution to this area.

For symptomatic cases, we used posterior median profiles reported by Hellewell et al. [55] of the probability of detecting infection against time since infection, with separate estimates for PCR and lateral flow tests (LFTs). The analysis used cycle threshold (Ct) data from repeat PCR testing of healthcare workers in the SAFER study [56], with infections confirmed by paired serology. The probability of detection by LFT was estimated given an assumption that an LFT would detect infections with a $Ct \leq 27$.

For asymptomatic cases, we assumed that the probability of asymptomatic individuals testing positive is equal to that of symptomatic individuals until the peak of infection, but then decays more rapidly, such that the probability of an asymptomatic individual testing positive at 6.7 days after the peak should equal the probability of a symptomatic individual testing positive at 10.5 days after the peak (corresponding with findings from Kissler et al. [53] who estimated an average duration of clearance of 10.5 days in symptomatic cases versus 6.7 days in asymptomatic cases) (figure 21).

The sensitivity of PCR tests when conditioned on having received a positive LFT result may differ from the sensitivity estimates of an independent PCR test. We assumed that individuals receiving a positive LFT result would be certain to return a positive result from the confirmatory PCR test.

## D.2. Test specificity

We assumed the specificity of PCR tests to be 100%, in line with the ONS UK COVID-19 Infection Survey indicating the specificity of the used PCR tests being in excess of 99.9% [57,58]). We assumed LFT specificity to be 99.68% [59]. Using LFTs to test entire year groups, false positives would be expected to occur relatively frequently.

## D.3. Model change log

We detail here notable parameter changes and additions to the previously presented network model [27].

### D.3.1. Isolation length

From 14 December 2020, the guidance from the UK government on the period of isolation for contacts of confirmed cases was reduced from 14 days to 10 days. The corresponding periods of isolation have been revised in the model.

### D.3.2. Infection risk for students awaiting return to university

For susceptibles not yet returned to the university, we computed a daily probability of infection to give a background prevalence of between 0.5% and 2% (with an infection duration of 16 days, across latent and infectious periods). We sampled the background prevalence in each simulation replicate from a Uniform (0.005, 0.02) distribution.

### D.3.3. Proportion of individuals who stayed in university accommodation between terms

Student surveys indicated that of the order of 10% of students intended to stay in their university accommodation after the end of the first academic term [48].

In each simulation replicate, we sampled the proportion independently for on-campus and off-campus residents from a Uniform (0.05,0.15) distribution, thus ensuring we included uncertainty associated with this quantity across our collection of simulations.

### D.3.4. Contact patterns

We applied the following two contact pattern changes to all but the baseline (no intervention) scenario: (i) society contacts did not occur (transmission risk therefore zero), with it assumed that all meetings would take place online; (ii) for on-campus resident students, we set a zero probability of a contact being made with an individual within the broader accommodation unit of the same floor or block of residence (thus outside the immediate household).

### D.3.5. Fraction of previous infecteds with PCR positive test result in the previous 90 days

In each simulation replicate, we sampled the fraction of previous infecteds who had returned a PCR positive test result in the previous 90 days from a uniform distribution, Uniform(0.02,0.05).

Individuals set as being present in accommodation prior to the start of the simulation entered the return testing procedure in an equivalent way to individuals with later arrival dates, with entry time determined by the relevant staggered student return strategy. For individuals from this group that became symptomatic and received a positive test result in the gap before their envisaged time to begin the return test process, they satisfied the condition of having had a positive PCR result within the previous 90 days and, as a consequence, no longer underwent the return test process.

### D.3.6. Assumptions for scenarios related to isolation status under staggered return and leaving return testing process

Returning students that have symptoms are by definition non-adherent to guidance. In this situation, for the household the returning student is joining, other adhering household members may enter household isolation. We assumed any such individuals entered isolation for the full 10-day period, irrespective of the date of symptom onset of the symptomatic individual.

In the scenario of a student completing the return testing procedure with negative results, and who would be entering a household that had household members in isolation due to the presence of a recently confirmed case, the student leaving the return test process would immediately enter household isolation.

## Appendix E. Additional information on the asymptomatic screening model

For this analysis, we used a layered network model of contacts between 15 000 simulated students, with one layer of household contacts and one of other-group contacts intended to simulate all out-of-household contact. We start the simulation with 100 infectious individuals, and run the model for 100 timesteps (notionally days). For each scenario, we plot the results of 100 replicates, each run on a newly generated network. Importantly: the particular parameters for this model have been chosen for a combination of plausibility and simplicity, and some are not well-founded in any particular dataset (we attempt to highlight these).

Half of the households were of 10 people, and half of 5 people (to simulate a cluster-flat arrangement in large halls, e.g. [60]). Other-group contacts are added in 3000 groups, with 5% of groups of size 40, 30% of size 10, 50% of size 5 and 15% of size 3—these values were chosen to simulate a range of activities, but are not well-founded in data. Results are not sensitive to small perturbations in these group sizes, but are sensitive to large changes in the overall amount of group contact. Within either household or other groups all individuals are assumed to have pairwise contact at all timesteps when the individuals are not isolating.

Disease progression and isolation are governed by a stochastic rate-based compartmental model in which individuals can be susceptible, exposed but not yet infectious, presymptomatically infectious, asymptotically infectious, symptomatically infectious, or recovered (and presumed immune). They can also be in these various states and self-isolating with their household. Individuals become exposed when one of their network contacts infects them—here household contacts have a 2.5% day$^{-1}$ probability of infecting each of their susceptible household members (note that this is independent of household size), and non-household contacts transmit with 1/10th this probability. These probabilities are increased by a factor of 1.5 when simulating a more-transmissible variant. These transmission figures have been chosen for simplicity and to plausibly reflect reasonable within-household attack rates. Where no other citation is given, rates of progression between disease states are round-number versions of the fitted parameters from [61]. Exposed individuals become presymptomatically or asymptomatically infectious at a rate of $0.33\,\text{day}^{-1}$ to give a mean 3-day latent period. Presymptomatically infectious individuals become symptomatic at a rate of $0.5\,\text{day}^{-1}$ to give a mean 2-day presymptomatic period. Symptomatically infectious people recover at a rate of $0.1\,\text{day}^{-1}$ to give a mean symptomatic infectious period of 10 days, a round-number version of the 9.5 days reported in [62]. We do not include hospitalization or death, as these events are very rare in the young-adult population. Half of infected individuals are assumed to develop symptoms, and half to remain asymptomatic (or non-test-seeking for some other reason). Asymptomatic individuals are infectious for the same mean total period of time as symptomatic individuals, and are equally infectious— predictably the effectiveness of asymptomatic screening is sensitive to this assumption.

Both symptomatic and asymptomatic testing are assumed to be perfect and rapid, returning results on the day of testing and giving neither false positives nor false negatives. Symptomatic individuals are assumed to immediately seek testing on the day symptoms develop. When an individual receives a positive test, they and their entire household are assumed to isolate perfectly from all non-household contacts, but continue to interact with household contacts as before. Non-household contacts of test-positives are traced and isolated with probability 0.5.

This model is an adaptation of a model originally written to model COVID-19 in Caribbean communities, available at https://github.com/SaraJakubiak/covid19-caribbean-educational-model— the majority of features within that model (including dynamically changing network, age-structure, etc.) are not used here. The adaptation of this code to the HE setting used to produce these results can be found at: https://github.com/magicicada/covid19-caribbean-educational-model/tree/manuscript-INI-HE-group.

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
