## [Peer Review File · Royal Society Open Science]

Review History

RSOS-210310.R0 (Original submission)

Review form: Reviewer 1

Is the manuscript scientifically sound in its present form?

Yes

Are the interpretations and conclusions justified by the results?

Yes

Is the language acceptable?

Yes

Do you have any ethical concerns with this paper?

No

Have you any concerns about statistical analyses in this paper?

No

Recommendation?

Major revision is needed (please make suggestions in comments)

Comments to the Author(s)

Paper Summary & Contribution

The focus of this paper is on leveraging modeling to inform policy decisions surrounding the pandemic-response and management of higher education in the UK. Specifically, the authors examine COVID-19 case data from the Fall 2020 semester to evaluate the effectiveness of various features of outbreak mitigation used by universities. The authors then use simulation modeling to determine the efficacy and value of return-to-campus interventions such as temporally staggered student arrivals. The main three findings highlighted by the authors include the following:

1. evidence of spillover transmission between university populations and the wider community in some, but not all, settings;
2. reductions in adherence to non-pharmaceutical interventions (NPIs) likely have a bigger impact than any marginal benefits from a staggered return to campus; and
3. the emergence of more transmissible variants reduces the effectiveness of mass asymptomatic testing.

Methodology

- The analysis shown in Figure 3 is interesting. I wonder if the authors could speak to what is driving some of the differences seen in these different regions? Could this be driven by differences across university testing / contact tracing / quarantine efforts? What about the concentration of restaurants / pubs / shopping versus universities that are near largely residential neighborhoods?
- For the staggered return to campus analysis, would it be possible to include an analysis of the resources required to accommodate certain return strategies? Since I would imagine this would be one of the key reasons universities decide to stagger student return. Testing and staffing requirements can be costly, and this could be an important factor that administrators consider when making these decisions.

Relevant Context

- It would be really helpful if the authors could add some related literature to place their own analysis and recommendations in the context of other work. There has been a flurry of research activity on COVID-19 policy over the past year and adding some context on whether this analysis is in agreement or disagreement with other work would be incredibly helpful. If the questions and recommendations here are completely novel, that would also be good to know.

Paper Flow

- The flow of the paper currently reads like a lab notebook with various sections for each type of model the authors built. I'm not sure if this is something that can be corrected due to the nature of the work, but perhaps presenting a roadmap in the introduction section that describes the logic behind why the following models will be presented would help the readers orient to what they are about to read. Otherwise, the lack of flow makes it difficult to get through.

Policy Recommendation

- Together, this body of work represents valuable thoughts on return-to-campus planning for university administrators. Despite the limitations of each of the models, this paper could be helpful in guiding administrators on what factors to consider when making these planning decisions.
- Something that could be improved is a more robust discussion of the costs associated with some of these interventions. Though it will undeniably be heterogeneous across universities, currently there is no discussion on the costs associated with some of these interventions, and that can often be a major deciding factor.

Minor Comments

- Please define epidemiological terms such as the secondary attack rate. This would make the paper much more accessible to a non-specialist audience.
- Consider presenting R-squared values in your regression tables.

Recommendation

- The paper addresses a timely question at the intersection of science, society, and policy. Though the policy recommendations made by the authors are not individually rigorous enough to be directly implemented, the models put together offer valuable insights for administrators who find themselves with this planning problem. I have included several recommendations on areas where the paper could be improved, and recommend a major revision so the authors might have a chance to make these changes. I wish the authors well and look forward to seeing an improved version of this paper in the future.

Decision letter (RSOS-210310.R0)

Dear Dr Tildesley

On behalf of the Editors, we are pleased to inform you that your Manuscript RSOS-210310 "SARS-COV-2 INFECTION IN UK UNIVERSITY STUDENTS: LESSONS FROM SEPTEMBER-DECEMBER 2020 AND MODELLING INSIGHTS FOR FUTURE STUDENT RETURN" has been accepted for publication in Royal Society Open Science subject to minor revision in accordance with the referees' reports. Please find the referees' comments along with any feedback from the Editors below my signature.

Please accept our apologies for the unusual delay in getting to this point: it has been exceptionally difficult to secure reviewers for the work - a state of affairs we very sorry for. However, we hope this decision is reassuring.

The Editors wanted to emphasise the following in preparing your revision:

- 1) While the nominal deadline of 7 days is usual for minor revisions, if you need an extension, this is absolutely fine - please contact me at the email address below, and we can push this back as required.
- 2) Please aim to address the main points outlined by the reviewer, in particular those relating to 'signposting' of the work for the reader - in other words, trying to clarify the aim and structure of the paper in the introduction, so it is clear what the paper is setting out to achieve.

3) The Editors are broadly supportive of the work (hence the decision) and don't consider any major re-working of the analysis to be needed, but agree with the reviewer that clarity of focus/message would be beneficial, especially given the policy implications of the work.

Please submit your revised manuscript and required files (see below) no later than 7 days from today's (ie 11-Jun-2021) date. Note: the ScholarOne system will 'lock' if submission of the revision is attempted 7 or more days after the deadline. If you do not think you will be able to meet this deadline please contact the editorial office immediately.

on behalf of Dr Rob Doubleday (Associate Editor) and Nick Pearce (Subject Editor)
openscience@royalsociety.org

Reviewer comments to Author:
Reviewer: 1
Comments to the Author(s)
Paper Summary & Contribution

The focus of this paper is on leveraging modeling to inform policy decisions surrounding the pandemic-response and management of higher education in the UK. Specifically, the authors examine COVID-19 case data from the Fall 2020 semester to evaluate the effectiveness of various features of outbreak mitigation used by universities. The authors then use simulation modeling to determine the efficacy and value of return-to-campus interventions such as temporally staggered student arrivals. The main three findings highlighted by the authors include the following:

1. evidence of spillover transmission between university populations and the wider community in some, but not all, settings;
2. reductions in adherence to non-pharmaceutical interventions (NPIs) likely have a bigger impact than any marginal benefits from a staggered return to campus; and
3. the emergence of more transmissible variants reduces the effectiveness of mass asymptomatic testing.

Methodology

- The analysis shown in Figure 3 is interesting. I wonder if the authors could speak to what is driving some of the differences seen in these different regions? Could this be driven by differences across university testing / contact tracing / quarantine efforts? What about the concentration of restaurants / pubs / shopping versus universities that are near largely residential neighborhoods?

- For the staggered return to campus analysis, would it be possible to include an analysis of the resources required to accommodate certain return strategies? Since I would imagine this would be one of the key reasons universities decide to stagger student return. Testing and staffing requirements can be costly, and this could be an important factor that administrators consider when making these decisions.

Relevant Context

- It would be really helpful if the authors could add some related literature to place their own analysis and recommendations in the context of other work. There has been a flurry of research activity on COVID-19 policy over the past year and adding some context on whether this analysis is in agreement or disagreement with other work would be incredibly helpful. If the questions and recommendations here are completely novel, that would also be good to know.

Paper Flow

- The flow of the paper currently reads like a lab notebook with various sections for each type of model the authors built. I'm not sure if this is something that can be corrected due to the nature of the work, but perhaps presenting a roadmap in the introduction section that describes the logic behind why the following models will be presented would help the readers orient to what they are about to read. Otherwise, the lack of flow makes it difficult to get through.

Policy Recommendation

- Together, this body of work represents valuable thoughts on return-to-campus planning for university administrators. Despite the limitations of each of the models, this paper could be helpful in guiding administrators on what factors to consider when making these planning decisions.

- Something that could be improved is a more robust discussion of the costs associated with some of these interventions. Though it will undeniably be heterogeneous across universities, currently there is no discussion on the costs associated with some of these interventions, and that can often be a major deciding factor.

Minor Comments

- Please define epidemiological terms such as the secondary attack rate. This would make the paper much more accessible to a non-specialist audience.

- Consider presenting R-squared values in your regression tables.

Recommendation

- The paper addresses a timely question at the intersection of science, society, and policy. Though the policy recommendations made by the authors are not individually rigorous enough to be directly implemented, the models put together offer valuable insights for administrators who find themselves with this planning problem. I have included several recommendations on areas where the paper could be improved, and recommend a major revision so the authors might have a chance to make these changes. I wish the authors well and look forward to seeing an improved version of this paper in the future.

===PREPARING YOUR MANUSCRIPT===

===PREPARING YOUR REVISION IN SCHOLARONE===

- An individual file of each figure (EPS or print-quality PDF preferred [either format should be produced directly from original creation package], or original software format).
 - An editable file of each table (.doc, .docx, .xls, .xlsx, or .csv).
 - An editable file of all figure and table captions.
- Note: you may upload the figure, table, and caption files in a single Zip folder.
- Any electronic supplementary material (ESM).
 - If you are requesting a discretionary waiver for the article processing charge, the waiver form must be included at this step.
 - If you are providing image files for potential cover images, please upload these at this step, and inform the editorial office you have done so. You must hold the copyright to any image provided.
 - A copy of your point-by-point response to referees and Editors. This will expedite the preparation of your proof.

- Ensure that your data access statement meets the requirements at <https://royalsociety.org/journals/authors/author-guidelines/#data>. You should ensure that you cite the dataset in your reference list. If you have deposited data etc in the Dryad repository, please only include the 'For publication' link at this stage. You should remove the 'For review' link.
- If you are requesting an article processing charge waiver, you must select the relevant waiver option (if requesting a discretionary waiver, the form should have been uploaded at Step 3 'File upload' above).
- If you have uploaded ESM files, please ensure you follow the guidance at <https://royalsociety.org/journals/authors/author-guidelines/#supplementary-material> to include a suitable title and informative caption. An example of appropriate titling and captioning may be found at https://figshare.com/articles/Table_S2_from_Is_there_a_trade-off_between_peak_performance_and_performance_breadth_across_temperatures_for_aerobic_scope_in_teleost_fishes_/3843624.

Author's Response to Decision Letter for (RSOS-210310.R0)

See Appendix A.

Decision letter (RSOS-210310.R1)

Dear Dr Tildesley,

It is a pleasure to accept your manuscript entitled "SARS-COV-2 INFECTION IN UK UNIVERSITY STUDENTS: LESSONS FROM SEPTEMBER-DECEMBER 2020 AND MODELLING

INSIGHTS FOR FUTURE STUDENT RETURN[™] in its current form for publication in Royal Society Open Science.

COVID-19 rapid publication process:

We are taking steps to expedite the publication of research relevant to the pandemic. If you wish, you can opt to have your paper published as soon as it is ready, rather than waiting for it to be published the scheduled Wednesday.

This means your paper will not be included in the weekly media round-up which the Society sends to journalists ahead of publication. However, it will still appear in the COVID-19 Publishing Collection which journalists will be directed to each week (<https://royalsocietypublishing.org/topic/special-collections/novel-coronavirus-outbreak>).

If you wish to have your paper considered for immediate publication, or to discuss further, please notify openscience_proofs@royalsociety.org and press@royalsociety.org when you respond to this email.

on behalf of Dr Rob Doubleday (Associate Editor) and Nick Pearce (Subject Editor)
openscience@royalsociety.org

Appendix A

We would like to thank the editor and the reviewer for the comments on our manuscript. Below we provide a response to the reviewer comments and indicate the changes that we have made to the manuscript.

Reviewer 1 comments and response (in bold):

Methodology

1 - The analysis shown in Figure 3 is interesting. I wonder if the authors could speak to what is driving some of the differences seen in these different regions? Could this be driven by differences across university testing / contact tracing / quarantine efforts? What about the concentration of restaurants / pubs / shopping versus universities that are near largely residential neighborhoods?

Unfortunately, the high levels of heterogeneity across university campuses and their surroundings make direct comparisons about the role of individual variables affecting transmission intractable with the data available. Some of this variability is illustrated in Table 4 and Figure 2 of the manuscript. Case studies of each of the selected university towns in Figure 3 is beyond the scope of this paper, as these were chosen as general illustrations of the types of spillover effects that might be observed.

From informal inspection, we agree with the reviewer that some contextual information, such as which Tier Level the population was assigned in late September and early October, and the magnitude of the student population relative to the wider community, weakly correlate with higher spillover signals. However, there are counterexamples to these trends as well, restricting their impact to the realm of speculation or to an in-depth analysis for an individual student body.

We completely agree with the reviewer that the efficacy and adherence of testing, tracing, and quarantine efforts are crucial to managing outbreaks, as later results in the manuscript also highlight. However, these aspects suffer from data sparseness, and are skewed by sampling and reporting biases. As such, we have not included these considerations in the analysis.

Additional text in the manuscript [in results part 2.4.2 after "interaction in different local authorities"]:

"Considerations such as the severity of imposed NPIs, magnitude of student body, and uptake and efficacy of testing, tracing, and quarantining measures likely all influence the overall results, but their individual contributions are not identifiable in this analysis."

2 - For the staggered return to campus analysis, would it be possible to include an analysis of the resources required to accommodate certain return strategies? Since I would imagine this would be one of the key reasons universities decide to stagger student return. Testing and staffing requirements can be costly, and this could be an important factor that administrators consider when making these decisions.

We thank the reviewer for raising this item. We agree that costs and logistical factors are an important consideration in the decision making process (please also see our response to point 6). That being said, in our analysis of staggered student return to university settings we are focused on the epidemiological consequences of staggering student return on SARS-CoV-2 transmission and isolation. We believe that the inclusion of resource considerations has the makings of a full research programme, requiring significant extensions to the modelling frameworks presented here.

We have amended the opening text in Section 3 to make our epidemiological study focus more prominent, recognise the factors the analysis does not account for and signpost to the reader that we elaborate on the study limitations in the Conclusions section:

“The intention of our modelling work was to focus purely on unpicking the epidemiological consequences of staggering student return on SARS-CoV-2 transmission and isolation. We acknowledge there are multiple factors that administrators must consider and there may be operational and/or resource reasons why a staggered return at Higher Education institutions is desired. These include ensuring that testing capacity is sufficient to meet demand, the monetary costs associated with the intervention (e.g. testing and staffing) and the educational needs of the students. Though the inclusion of these considerations is beyond the scope of our study, they are important constituents of a multi-faceted decision making process and we provide an expanded discussion in the Conclusions section.”

Relevant Context

3 - It would be really helpful if the authors could add some related literature to place their own analysis and recommendations in the context of other work. There has been a flurry of research activity on COVID-19 policy over the past year and adding some context on whether this analysis is in agreement or disagreement with other work would be incredibly helpful. If the questions and recommendations here are completely novel, that would also be good to know.

We agree that this is important and have added in additional paragraphs in the conclusion which compare our results to other literature:

“The mass migration of students at the beginning and end of academic terms, their unique living arrangements during term time and unique patterns of social mixing, make them an important population for the spread of infectious respiratory illnesses. Despite this, prior to the COVID-19 pandemic, there was little data collected on outbreaks of infectious disease at universities (although one such dataset collected between October 2007 and mid-February 2008 has now been published in 2021 \cite{Eames2021}) and university students were an understudied population. Therefore, at the start of the pandemic there was a limited evidence base to support policy decisions around universities. Our study brings together expertise from multiple research groups and presents results from multiple statistical and modelling analyses and provides new understanding on infectious disease outbreaks at universities and how these could be mitigated.

An important finding of our study is that adherence to NPIs is likely to have more impact than staggering the return of students to university. Survey data suggest that in the autumn term of 2020, students generally did have high adherence to NPIs; an Office for National Statistics (ONS) survey found high adherence (90\%) to social distancing across multiple universities \cite{ONSsurvey}. In addition, a survey of University of Bristol students found that 99\% of students self-isolated after testing positive for COVID-19 and the majority of survey participants reported low contact numbers \cite{Nixon2021}. However, there was heterogeneity in adherence, with some students reporting many contacts and with only 61\% of students with cardinal COVID-19 symptoms self-isolating \cite{Nixon2021}. In future, it will be important that students maintain their high levels of adherence and to ensure they have sufficient resources to allow them to do so.

Several of the scenarios presented here have considered the frequency of asymptomatic screening at universities. This has been explored in other modelling studies, for example \cite{Lopman2021} found that monthly screening can reduce cumulative incidence by 59\% and weekly screening by 87\%. We found that increasing the frequency of asymptomatic screening is likely to be important in the presence of a more transmissible SARS CoV-2 variant, with cases only being able to be maintained below 1200 (mean cumulative over 100 days) when testing occurs every 3 days (in a population of 25,000). This finding corroborates with a study that used an agent based model to simulate COVID-19 transmission at the University of California San Diego, where larger outbreaks resulted in a maximum outbreak size of 158 when asymptomatic screening occurred monthly and 7 when it occurred twice weekly \cite{Goyal2021}, but with a much lower impact seen on the average outbreak size when increasing from monthly to twice weekly testing, ranging from 1.9 to 1.1 respectively. Brooks Pollock et al. \cite{Brooks-Pollock2020} also found that mass testing was more effective for higher values of the reproduction number. This highlights the importance of reassessing control measures under different variants.

We have focused here on COVID-19 risks and mitigation strategies for when students return to university and during the university term itself, however, we have covered little on the risk of transmission from infected students to private homes at the end of term. Previous modelling work suggests that in an unvaccinated population, an infectious student would on average generate just less than one secondary within-household infection, but this is dependent on the prevalence in the student population at the time of departure \cite{Harper2021}. Although it is expected that vaccination will reduce the impact of students returning to private homes at the end of term, the UK vaccination program is ongoing and there are particular spatial areas and demographic groups where low uptake is expected \cite{deFigueiredo2020.12.17.20248382}, suggesting that this still may be an important question to consider in future.”

We have also added in section 3.5.3 “In addition, the model did not include a reduction in the risk of transmission occurring over contacts due to face covering use or social distancing, however other work \cite{Hambridge2021} suggests that if such measures are in place in a university setting and/or if there are moderate levels of immunity, the impact of testing is less prominent, highlighting the importance of considering testing in the context of other measures.”

In the conclusion we also have added in “For example, to sustain a regular testing regime at universities under financial, logistical, or structural constraints, mathematical modelling suggests that pooling RT-qPCR testing may be a cost-effective method, although this may come with additional caveats resulting from the associated reduction in sensitivity (when cases are not detected) and sensitivity (when students self-isolate but are not infected) \cite{Hemani2021}.”

Paper Flow

4 - The flow of the paper currently reads like a lab notebook with various sections for each type of model the authors built. I’m not sure if this is something that can be corrected due to the nature of the work, but perhaps presenting a roadmap in the introduction section that describes the logic behind why the following models will be presented would help the readers orient to what they are about to read. Otherwise, the lack of flow makes it difficult to get through.

We thank the reviewer for this comment. We have gone through the paper and reorganised the introduction and conclusion sections to include the core results and a roadmap for the paper (with a figure) has been included in the introduction. We believe that this significantly helps to improve the flow of the paper.

5 - Together, this body of work represents valuable thoughts on return-to-campus planning for university administrators. Despite the limitations of each of the models, this paper could be helpful in guiding administrators on what factors to consider when making these planning decisions.

We would like to thank the reviewer for this comment. We agree that this work can help to inform administrators regarding future planning decisions and this group are currently working towards an approach to disseminate these findings to the wider higher education sector.

6 - Something that could be improved is a more robust discussion of the costs associated with some of these interventions. Though it will undeniably be heterogeneous across universities, currently there is no discussion on the costs associated with some of these interventions, and that can often be a major deciding factor.

We are in agreement with the reviewer that resource considerations associated with public health interventions can be determining decision factors in the programmes undertaken. We think that the costs of the interventions should be viewed in a broader context. The resources used by the universities for testing are difficult to separate from the resources used for teaching and research. As an example, many of the testing schemes so far were run either as research projects or with heavy research components. We should also not lose sight of the primary purposes of universities, which are to deliver teaching and research. Closing a university is, of course, disruptive to both of these. The possible advantages of the implementation of measures, such as staggering, on COVID safety must always be balanced against the cost in terms of the disruption of the teaching and research activity. Indeed any staggering strategy should be developed in tandem with the necessary changes to the teaching schedules. This is an important future direction of study, which we now expand upon in the Conclusions.

With this in mind, we have added the following text into the discussion section of the manuscript and added additional references that have considered the non-COVID impact of the pandemic:

“We recognise there are prominent factors that we have not addressed here as we have focused directly on transmission dynamics, yet should be considered while viewing our results in a broader context. One important future research direction is to consider the non-COVID impact of intervention measures. The majority of work to date on COVID-19 has focused upon developing intervention policies that seek to minimise the overall number of cases, hospital admissions or deaths. However, it is important to acknowledge that any control policy that may reduce transmission also has an impact in terms of monetary cost, non-COVID health, mental health and well being. An extension to this work could focus upon

assessing the direct monetary cost of intervention policies as well as the logistical and operational constraints associated with such policies. Additionally, in higher educational settings, it is important to consider any impact on teaching and examination schedules as well as mental health and well being of students. The models considered here allow for an estimate of the different resources used by the different control strategies. In order to determine an optimal intervention, it is crucial to establish the objective of any control policy, noting that the objective may not be generalisable across all higher education establishments. Once an objective is appropriately defined, any modelling can be specifically tailored to maximise the robustness of any advice offered.

Furthermore, a growing picture is just beginning to emerge on the prevalence of, and risk factors for, 'long COVID' symptoms and health complications following coronavirus (COVID-19) infection. An initial set of early experimental results collected by the ONS indicates around 1 in 5 respondents testing positive for COVID-19 exhibit symptoms for a period of 5 weeks or longer, and around 1 in 10 respondents testing positive for COVID-19 exhibit symptoms for a period of 12 weeks or more.

We recognise that the current university closures may have significant impact upon student mental health and well-being -- across multiple surveys collecting information on how the COVID-19 pandemic has affected the mental health of students, a consistent outcome was above 50% of respondents expressing that their well-being and mental health had become worse. In addition, we hope that the ongoing vaccine rollout will provide a level of protection for those most vulnerable to severe outcomes, which in turn may alleviate risks associated with possible student to community spread."

Minor Comments

7 - Please define epidemiological terms such as the secondary attack rate. This would make the paper much more accessible to a non-specialist audience.

We agree that clarification of terminology would make this work more accessible to our target audience. We have added a definition for secondary attack rate in the introduction to Section 2.2 (lines 159-161):

"We refer to the secondary attack rate (SAR) in a sub-population (e.g. household, hall of residence) as the probability that a member of the sub-population is infected following infection of one sub-population member.

We have also defined a number of other epidemiological terms; age-stratified (Section 2, line 99), prevalence (see Section 2.1.1, line 111), incidence (Section 2.3, lines 246-7), generation time (Section 2.3, lines 265-6), SIR model (Section 3.2, lines 423-435) and model stochasticity (Section 3.3, lines 456). In addition we have included a brief

description of the infectious stages in the models used to explore mass asymptomatic testing (Section 3.3, lines 459-61) and provided interpretation for the terms test sensitivity and specificity (Section 3.3, lines 534-5). Where possible we have rewritten text to avoid using technical terms (e.g. serology, heterogeneity). Abbreviations for software packages which may be unfamiliar have been expanded (Section 2.2.2, lines 183-5) and we provided a brief interpretation of violin plots (Section 3.3.1, lines 509-10).

8 - Consider presenting R-squared values in your regression tables.

Since logistic regression models are used, rather than linear, a pseudo R-squared would need to be calculated. McFadden's pseudo R-squared could be used, however the interpretation of this differs from a linear regression R-squared, with values of 0.2 to 0.4 representing an excellent fit. There is a risk that if the pseudo R-squared were included this may lead to readers interpreting it as a normal R-squared value.

We have included the following paper summary with the resubmission:

We present analyses on SARS-CoV-2 transmission in UK higher education settings with two main objectives. Firstly, we study the observed patterns of SARS-CoV-2 in universities from September to December 2020. We found that the student-community transmission relationships were not the same in all local areas. Secondly, using mathematical models we assess the impact of potential control measures for when students return. Staggering future returns has limited epidemiological benefits and we found a reduced ability of mass testing to suppress case numbers in the presence of more transmissible variants. Strong adherence of students to control measures appears crucial to their effectiveness.